# Neural precursor cells rescue symptoms of Rett syndrome by activation of the Interferon γ pathway

Angelisa Frasca [1,8], Federica Miramondi [1,8], Erica Butti[2], Marzia Indrigo[3], Maria Balbontin Arenas [1], Francesca M Postogna[1], Arianna Piffer[1,5], Francesco Bedogni[3,6], Lara Pizzamiglio[1,7], Clara Cambria[1], Ugo Borello [4], Flavia Antonucci [1], Gianvito Martino[2] & Nicoletta Landsberger [1,3 ✉]

## Abstract

The beneficial effects of Neural Precursor Cell (NPC) transplantation in several neurological disorders are well established and they are generally mediated by the secretion of immunomodulatory and neurotrophic molecules. We therefore investigated whether Rett syndrome (RTT), that represents the first cause of severe intellectual disability in girls, might benefit from NPC-based therapy. Using in vitro co-cultures, we demonstrate that, by sensing the pathological context, NPC-secreted factors induce the recovery of morphological and synaptic defects typical of *Mecp2* deficient neurons. In vivo, we prove that intracerebral transplantation of NPCs in RTT mice significantly ameliorates neurological functions. To uncover the molecular mechanisms underpinning the mediated benefic effects, we analyzed the transcriptional profile of the cerebellum of transplanted animals, disclosing the possible involvement of the Interferon γ (IFNγ) pathway. Accordingly, we report the capacity of IFNγ to rescue synaptic defects, as well as motor and cognitive alterations in *Mecp2* deficient models, thereby suggesting this molecular pathway as a potential therapeutic target for RTT.

**Keywords** Cytokine; Mecp2; Neurodevelopmental Disease; Stem Cells; Synapses
**Subject Categories** Genetics, Gene Therapy & Genetic Disease; Neuroscience

## Introduction

Mutations in the X-linked *MECP2* gene, encoding the methyl-CpG binding protein 2 (MeCP2), cause a broad spectrum of neuropsychiatric diseases including Rett syndrome (RTT), an early-onset neurodevelopmental disorder that represents the most common genetic cause of severe intellectual disability in girls worldwide (Chahrour and Zoghbi, 2007). RTT patients are diagnosed on the basis of clinical signs, which include a period of regression together with the following four main criteria: loss of spoken language, loss of purposeful hand movements, gait abnormalities and the presence of hand stereotypies. Other clinical symptoms, such as cognitive disabilities, impairments in ambulation, breathing abnormalities and seizures, vary in frequency and severity and emerge during the disease progression (Neul et al, 2010).

Both cellular and animal models contributed to expand our knowledge on RTT and MeCP2 functions. Studies on neurons derived from either transgenic mice or human induced pluripotent stem cells (hiPSCs) highlighted morphological and functional alterations typically associated with *MECP2* mutations. Indeed, RTT neurons display reduced soma size and dendritic arborization, spine dysgenesis and decreased synapses' number as well as altered calcium signaling and excitatory/inhibitory balance (Fukuda et al, 2005; Dani et al, 2005; Chapleau et al, 2009; Belichenko et al, 2009; Ananiev et al, 2011; Perego et al, 2022). RTT symptoms are well recapitulated by animal models mutated in *Mecp2* (Frasca et al, 2023). Although RTT mainly affects girls, male *Mecp2* knock-out (*Mecp2* KO or null) mice remain largely used in basic research and pre-clinical studies, instead of female heterozygous mice. This is due to the fully penetrant phenotype of KO mice and the fact that female heterozygous animals manifest milder and delayed symptoms with a higher variability compared to the hemizygous model, because of random X-chromosome inactivation.

RTT brains are characterized by severe neurobiological changes, which are recapitulated by *Mecp2* null mice. Although there is no neurodegeneration, we recently reported that *Mecp2* deficiency causes a delay in brain growth (Carli et al, 2023); furthermore, several studies described structural and functional abnormalities mainly affecting neurons and with brain region specificity (Armstrong, 2005). These alterations are paralleled and caused by molecular abnormalities (Scaramuzza et al, 2021), and in line with the role of MeCP2 as transcriptional regulator, its loss of function causes subtle but widespread gene deregulation (Pacheco et al, 2017; Gogliotti et al, 2018; Sanfeliu et al, 2019). However, RTT does

[1]Department of Medical Biotechnology and Translational Medicine, University of Milan, Segrate, Milan I-20054, Italy. [2]Neuroimmunology Unit, Division of Neuroscience, IRCCS San Raffaele Scientific Institute, Milan I-20132, Italy. [3]San Raffaele Rett Research Unit, Neuroscience Division, IRCCS San Raffaele Scientific Institute, Milan I-20132, Italy. [4]Cellular and Developmental Biology Unit, Department of Biology, University of Pisa, I-56127 Pisa, Italy. [5]Present address: Department of Oncology and Children's Research Center, University Children's Hospital Zurich, Zurich, Switzerland. [6]Present address: Neuroscience and Mental Health Innovation Institute (NMHII), Cardiff University School of Medicine, Cardiff CF24 4HQ, UK. [7]Present address: Institut de Biologie de l'École Normale Supérieure (IBENS), Ecole Normale Supérieure, Université PSL, CNRS, INSERM, F-75005 Paris, France. [8]These authors contributed equally as first authors: Angelisa Frasca, Federica Miramondi. ✉E-mail: nicoletta.landsberger@unimi.it

not spare glial cells that in fact are characterized by defective gene expression, morphology, and functionality (Ballas et al, 2009; Lioy et al, 2011; Delépine et al, 2015; Pacheco et al, 2017; Albizzati et al, 2022). Consequently, *Mecp2* null astrocytes provide limited support to neuronal maturation and synapse formation (Albizzati et al, 2024; Sun et al, 2023). The scenario is further complicated by the presence of metabolic and mitochondrial abnormalities (Kyle et al, 2018) and a dysregulation of immune response (Theoharides et al, 2015; Pecorelli et al, 2020), which likely play an active role in the generation and/or maintenance of RTT phenotypes. Evidence associating MeCP2 with the immune system is increasing, and cytokine dysregulation was described in RTT patients (Leoncini et al, 2015; De Felice et al, 2016; Byiers et al, 2019). Such a wide spectrum of symptoms challenges the identification of effective pharmacological therapies for RTT. Accordingly, despite several pre-clinical studies, success in clinical trials has been so far relatively poor (Frasca et al, 2023; Palmieri et al, 2023). However, very recently, Trofinetide, the synthetic version of the tripeptide glycine-proline-glutamate of IGF-1, was approved by FDA for RTT (NCT04181723). Furthermore, two phase I/II clinical trials for gene therapy in RTT were recently authorized (Palmieri et al, 2023).

Considering all above, we tested the therapeutic potential of a cell therapy, based on adult Neural Precursor Cells (NPCs). NPCs constitute a class of multipotent stem cells, which have been widely studied for their ability to improve neuropathological signs in many neurodegenerative diseases characterized by spatially circumscribed CNS lesions, such as Parkinson's disease and Huntington's chorea, and also in multiple sclerosis, ischemia, and spinal cord injury (Lindvall and Kokaia, 2010). The safety of applying NPCs to patients affected by multiple sclerosis was recently approved (Genchi et al, 2023).

Although they are initially expected to act by replacing damaged cells, NPCs can also protect the pathological brain through alternative mechanisms, involving the interaction of NPCs with resident neural and immune cells. NPCs can adapt their fate and functions to the diseased CNS and modulate astrocytes, microglia, and inflammatory cells in response to pathological processes through paracrine mechanisms (bystander effect). By releasing specific molecules (neurotrophic factors, reactive species, binding proteins, purines, or cytokines) and forming a close network that persists after administration, NPCs exert immunomodulatory or neuroprotective functions (Kokaia et al, 2012; Bacigaluppi et al, 2016; De Feo et al, 2017).

Other stem cells, such as mesenchymal cells, were widely used with therapeutic purposes in the field of neurodevelopmental and neurocognitive disorders, and a clinical trial on a small group of RTT patients treated with NPCs was reported on a Chinese journal (Liu et al, 2013; Nabetani et al, 2023). However, to the best of our knowledge, no study reports exhaustive results supporting the efficacy of NPCs in RTT. In this study, we tested our hypothesis that NPCs might exert positive effects in RTT models. In vitro, we prove that RTT neurons induce NPCs to secrete factors able to recover morphological and synaptic defects of null and heterozygous neurons. In vivo, we demonstrate that NPCs ameliorate many RTT-like symptoms. Activation of the Interferon gamma–pathway (IFNγ) emerges as an involved mechanism pointing to this cytokine as a novel and promising therapeutic target for RTT.

# Results

## By sensing the pathological context, NPCs secrete factors which promote neuronal maturation of *Mecp2* KO primary neurons

To determine whether NPCs exert beneficial effects in *Mecp2* deficient neurons, we initially used an in vitro system in which NPCs were seeded on transwell inserts and cultured with neurons from DIV0 to the end of the experiment (DIV7 for the analysis of dendritic branching and length or DIV14 for the analysis of synaptic puncta density) (Fig. 1A). This co-culture allowed to assess the paracrine effects exerted by NPCs in agreement with their well-known bystander mechanism (Kokaia et al, 2012). As controls we included WT and KO neurons cultured with NIH3T3 fibroblasts, in addition to neurons cultured alone. Analysis of dendrites reported the ability of NPC-secreted factors to rescue dendritic complexity and length in KO neurons (Fig. 1B–D). In fact, while a reduction in the number of intersections between dendrites and concentric circles in Sholl analysis was measured in both KO neurons and KO neurons cultured with fibroblasts (Fig. 1B,C), a significant increase was observed in KO neurons treated with NPCs (Fig. 1C). Accordingly, by analyzing the number of total intersections regardless of distance from the soma, we found that the significant reduction in KO neurons was rescued when they were exposed to NPCs (Fig. EV1B). A similar effect was obtained in WT neurons (Fig. EV1A,B), demonstrating the broad trophic effect of NPCs.

Importantly, NPC treatment also reverted the already described defects in synaptic puncta density affecting KO neurons (Frasca et al, 2020). Indeed, by immunostaining for Synapsin1/2 and Shank2, we proved the ability of NPCs to rescue both alterations in KO neurons, while NIH3T3 cells revealed ineffective (Fig. 1E–H). In line with these effects and the established role of Brain Derived Neurotrophic Factor (BDNF) on synaptic plasticity, WB analysis indicated the already reported reduction of Bdnf in KO neurons (Chang et al, 2006), that was recovered by NPC treatment (Fig. 1I). Interestingly, NPCs increased the neurotrophin levels also in WT neurons confirming their intrinsic trophic support.

By analyzing *Mecp2* heterozygous (Het) neurons, that better recapitulate the human disorder, we confirmed the ability of NPC-secreted molecules to increase the number of pre-synaptic puncta, without however completely rescuing the defect (Fig. 1J,K). Interestingly, a significant reduction of the number of pre-synaptic puncta was found without observing a bimodal data distribution, indicating the already proved occurrence of non-cell autonomous mechanisms between neurons that express either the *Mecp2* wild type or null allele (Johnson et al, 2017; Belichenko et al, 2009). At the functional level, we analyzed spontaneous excitatory postsynaptic events (mEPSCs) by whole-cell patch-clamp recordings, revealing increased mEPSCs frequency in Het cultures with respect to WT neurons. This alteration in frequency, that can be ascribed to several mechanisms, including an attempt of the system to compensate for the reduced pre-synaptic puncta, was fully rescued by NPCs but not by NIH3T3 cells. In addition, NPC secreted factors increased the mEPSC amplitude of Het neurons, which showed a slight and not significant reduction compared to WT cells ($p = 0.08$) (Fig. EV1C,D).

From a therapeutic perspective, we found interesting to test whether NPCs require Mecp2 to exert their therapeutic functions.

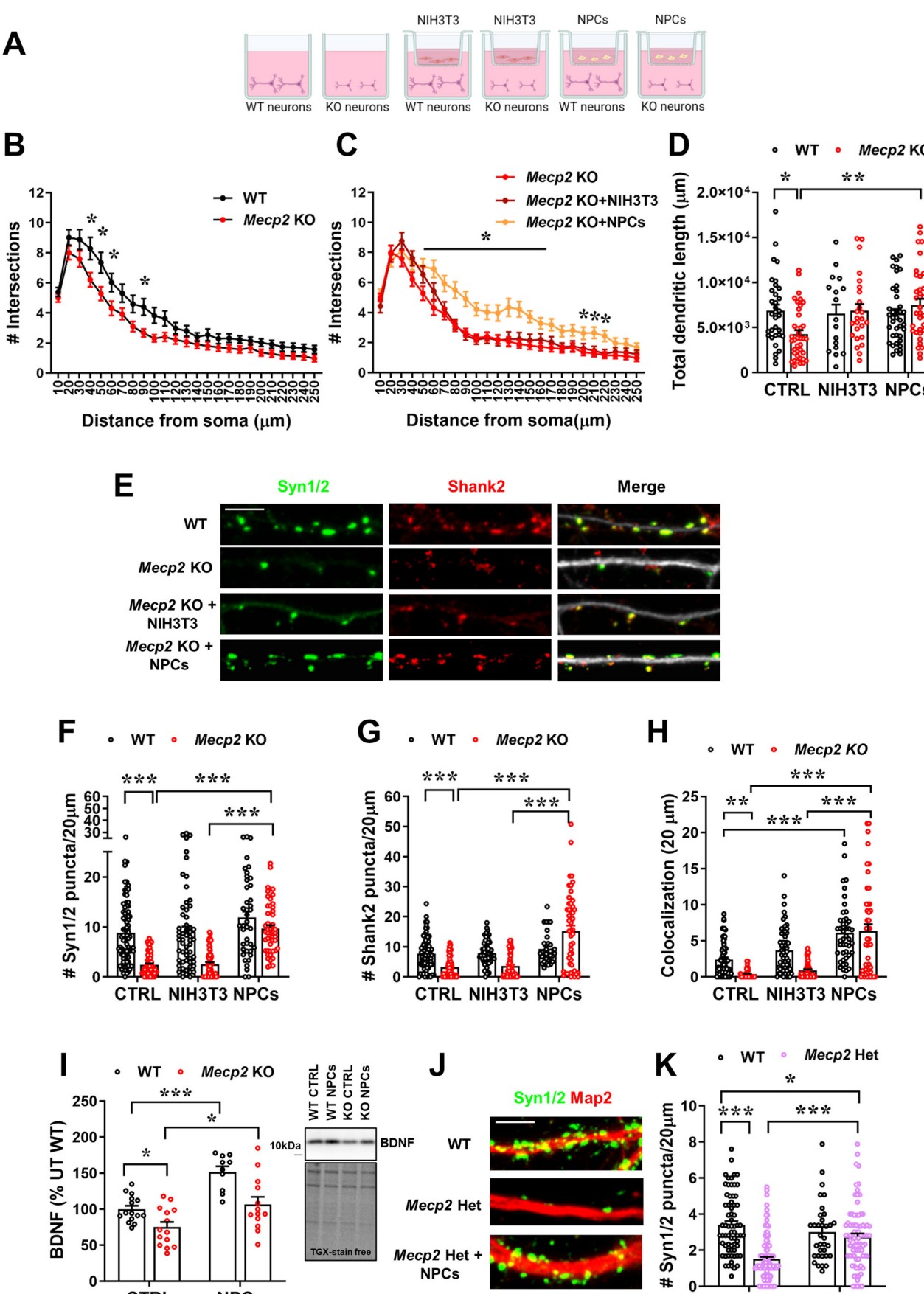

◀ **Figure 1. NPC treatment rescues dendritic branching and synaptic defects in *Mecp2* deficient neurons.**

(A) The cartoon depicts the experimental setting, in which NPCs or NIH3T3 were seeded on transwell inserts, which were transferred on WT or KO neurons at DIV0 until the end of the experiment (DIV7 for (B–D); DIV14 for (E–K)). (B, C) The graphs show the number of intersections calculated by Sholl analysis in WT and KO neurons (B), and in KO neurons cultured alone or with NIH3T3 or NPCs (C). In (B), *$p = 0.0030$ at 40 μm; $p = 0.033$ at 50 μm; $p = 0.0348$ at 60 μm; $p = 0.0317$ at 90 μm. In (C), *$p = 0.0018$ at 50 μm; $p < 0.0001$ at 60 μm; $p = 0.0003$ at 70 μm; $p < 0.0001$ at 80 μm; $p < 0.0001$ at 90 μm; $p = 0.0007$ at 100 μm; $p = 0.0049$ at 110 μm; $p = 0.0021$ at 120 μm; $p < 0.0001$ at 130 μm; $p < 0.0001$ at 140 μm; $p = 0.0007$ at 150 μm; $p = 0.0071$ at 160 μm; $p = 0.0082$ at 170 μm; $p = 0.0385$ at 200 μm; $p = 0.0224$ at 210 μm; $p = 0.0379$ at 220 μm. (D) The histogram indicates the total dendritic length calculated by NeuronJ in WT and KO neurons cultured alone (CTRL) or with NIH3T3 or NPCs. Data are reported as mean ± SEM. *$p = 0.0259$ WT *vs* KO; **$p = 0.0018$ KO *vs* KO+NPCs by two-way ANOVA, followed by Tukey post-hoc test. In (C) and (D), $n = 33$ WT, $n = 17$ WT + NIH3T3; $n = 39$ WT + NPC; $n = 39$ KO; $n = 26$ KO + NIH3T3; $n = 32$ KO+NPCs. Neurons derived from at least 3 different mice/genotype. (E) Representative images of the immunostaining for Synapsin1/2 (Syn1/2; green), Shank2 (red) and their merge with Map2 of WT and KO neurons (DIV14) cultured alone or co-cultured with NIH3T3 or NPCs. Scale bar = 5 μm. (F–H) Histograms indicate the mean ± SEM of number of Synapsin1/2 and Shank2 puncta in 20 μm (F, G) and of colocalized puncta (H) of WT and KO neurons cultured alone (CTRL) or co-cultured with NPCs or NIH3T3. Data were analyzed by two-way ANOVA followed by Tukey post-hoc test. In (F), ***$p < 0.0001$ WT *vs* KO; $p < 0.0001$ KO *vs* KO+NPCs; $p < 0.0001$ KO + NIH3T3 *vs* KO+NPCs. In (G), ***$p = 0.0003$ WT *vs* KO; $p < 0.0001$ KO *vs* KO+NPCs; $p < 0.0001$ KO + NIH3T3 *vs* KO+NPCs. In (H), ***$p = 0.0055$ WT *vs* KO; $p < 0.0001$ KO *vs* KO+NPCs; $p < 0.0001$ KO + NIH3T3 *vs* KO+NPCs. $n = 78$ WT, $n = 58$ WT + NIH3T3, $n = 41$ WT+NPCs, $n = 68$ KO, $n = 59$ KO + NIH3T3, $n = 49$ KO+NPCs. (I) The histogram represents the mean ± SEM of the protein levels of the mature form of BDNF in WT and KO neurons cultured alone or co-cultured with NPCs (from DIV0 to DIV14) and expressed as percentage of WT neurons. *$p = 0.0368$ WT *vs* KO; *$p = 0.0093$ KO *vs* KO+NPCs; ***$p < 0.0001$ WT *vs* WT+NPCs by two-way ANOVA followed by Sidak post-hoc test. $n = 15$ WT, $n = 10$ WT+NPCs, $n = 14$ KO, $n = 13$ KO +NPCs. Representative bands of BDNF and the corresponding lanes of TGX-stain free gel are reported. (J) Representative images of the immunostaining for Synapsin1/2 (Syn1/2; green) and Map2 (red) of WT and Het neurons (DIV14) cultured alone or co-cultured with NPCs. Scale bar = 5 μm. (K) The histogram indicates the mean ± SEM of Synapsin1/2 puncta density of WT and Het neurons cultured alone (CTRL) or co-cultured with NPCs. *$p = 0.0408$ WT *vs* Het+NPCs; ***$p < 0.001$ WT *vs* Het; ***$p < 0.001$ Het *vs* Het+NPCs by two-way ANOVA followed by Tukey post-hoc test; $n = 68$ WT, $n = 34$ WT+NPCs, $n = 109$ Het, $n = 83$ Het+NPCs. Neurons derived from at least 6 mice/genotype and 3 independent experiments were analyzed. Source data are available online for this figure.

Using the same experimental setting (Fig. 2A), we found that the lack of Mecp2 does not alter the ability of NPCs to correct synaptic alterations; in fact, WT and KO NPCs are equally effective in increasing pre- and post-synaptic densities, and their colocalization, in KO neurons (Fig. 2B–E).

To explore the hypothesis that NPCs secrete factors depending on the specific environment (Drago et al, 2013), we analyzed the synaptic phenotype in KO neurons treated for 24 h (DIV13-DIV14) with Conditioned Medium (CM) collected from the following co-cultures: WT neurons and NPCs (CM NPC^WT), KO neurons and NPCs (CM NPC^KO), KO neurons and NIH3T3 (CM NIH3T3^KO). KO and WT neurons cultured alone were used as controls (UT). Interestingly, the previously assessed synaptic defects that typically affect RTT neurons were exclusively rescued by the CM derived from NPCs co-cultured with KO neurons, indicating the capacity of the stem cells to sense the pathological environment and adapt their secretome in accordance with it, thereby releasing beneficial factors for RTT neurons (Fig. 3A–D).

## NPC transplantation ameliorates RTT-like symptoms in *Mecp2* deficient mice

To assess whether NPCs could improve RTT-related symptoms, 10 × 10^6 cells were transplanted by intra-cisterna magna (i.c.m.) injection in symptomatic (P45-47) *Mecp2* KO mice, and WT littermates as control. Survival and behavioral parameters, which correlate with the severity of the disease (Guy et al, 2007), were analyzed by a researcher blind to the treatment and genotype, and data compared with PBS-injected WT/KO mice (Fig. 4A). Kaplan–Meyer survival analysis indicated an initial shift in mortality, without modifying the two curves; notably, this shift was sufficient to cause a significant difference in the lifespan between NPC- and PBS-treated KO mice (Fig. 4B). A well-defined scoring system was used to measure the severity of RTT-like symptoms in a period ranging between 5 days before and 16 days after transplant (Guy et al, 2007). Evolution of symptoms was

graphically represented through a cumulative plot and a heatmap (Figs. 4C,D and EV2A–E), which show that NPCs induced a widespread recovery of symptoms in KO mice starting roughly 10 days after transplantation. To investigate NPCs' effectiveness in reverting neurological defects, animals were tested for their motor and cognitive functions. The rotarod test was used to assess both motor learning, comparing the latency to fall between the first trial with the subsequent trials, and motor coordination, analyzing the latency to fall in the last trial. Importantly, PBS-treated KO mice manifested impaired motor learning, which was significantly ameliorated by NPCs, as indicated by the significant difference between the time spent on the rod in the 3rd and 1st trial (Fig. 4E). As expected, WT animals (independently from the treatment) showed the appropriate motor learning. Analysis of motor coordination proved that the difference between WT and KO mice was completely rescued by NPCs (Fig. 4F). Furthermore, by performing the Novel Object Recognition (NOR) test, that evaluates the difference in the exploration time of a novel and a familiar object (expressed as Discrimination Index; D.I.), we reported that the significant defect in KO animals was no more present after NPC transplantation (Fig. 4G). The amelioration observed in the NOR test was purely cognitive and not influenced by mobility, since the impairment in the distance travelled inside the arena was not affected by NPCs (Figs. 4H and EV2F).

To characterize the molecular mechanisms by which transplanted NPCs exert positive effects in KO mice, we investigated NPC grafting, localization and differentiation. GFP staining revealed that most transplanted cells distributed posteriorly, in an area proximal to subarachnoid space of cerebellum, fourth ventricle and brainstem, in accordance with the site of injection (Fig. 5A,B). 10 days after transplantation we estimated that an average of 200,000 GFP+-NPCs persisted in the KO brain, which correspond to about 2% of the transplanted cells. Grafted NPCs expressed the stem cell marker Nestin, and most of them were positive for the astrocytic marker GFAP. However, the lack of a ramified astrocytic morphology suggested that these cells maintain an immature

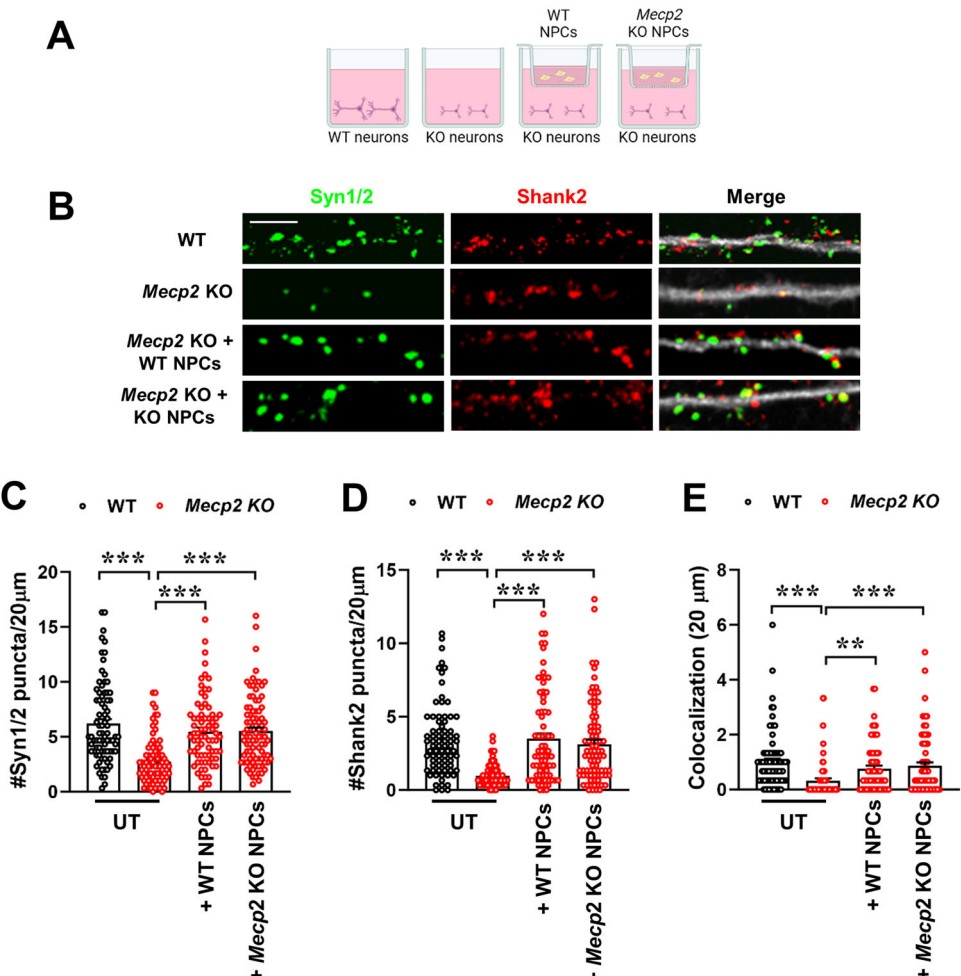

**Figure 2. Beneficial effects of NPCs do not require Mecp2.**

(A) The cartoon depicts the experimental setting of the co-cultures. (B) Representative images of the immunostaining for Synapsin1/2 (Syn1/2; green), Shank2 (red) and the merge with Map2 (white) of neurons (at DIV14). Scale bar = 5 µm. (C–E) The graphs show the mean ± SEM of number of Synapsin1/2 and Shank2 puncta in 20 µm (C, D) and of colocalized puncta (E) of WT and KO neurons cultured alone (UT) or co-cultured with WT NPCs or *Mecp2* KO NPCs. In (C) and (D): ***$p < 0.0001$ WT *vs* KO by unpaired t test; ***$p < 0.0001$ KO *vs* KO + WT NPCs, ***$p < 0.0001$ KO *vs* KO + KO NPCs by one-way ANOVA, followed by Dunnett's post-hoc test. In (E): ***$p < 0.0001$ WT *vs* KO by unpaired t test; **$p = 0.0076$ KO *vs* KO + WT NPCs, ***$p = 0.0003$ KO *vs* KO + KO NPCs by one-way ANOVA, followed by Dunnett's post-hoc test. $n = 84$ WT, $n = 83$ KO, $n = 77$ KO + WT NPCs, $n = 90$ KO + KO NPCs. Data derived from 6 mice/genotype and from 2 independent experiments. Source data are available online for this figure.

phenotype, probably committed to the astrocytic lineage (Fig. 5C). Thus, we hypothesize that, similarly to the in vitro observations, transplanted NPCs induce their beneficial action by sensing the pathological context and releasing soluble factors, therefore through a bystander mechanism.

Finally, we assessed NPC grafting and efficacy in P180 *Mecp2* Het mice, when symptoms are comparable to those observed in P45 KO males (Carli et al, 2023). Immunofluorescence analysis of GFP⁺ NPCs, performed 10 days after transplantation, revealed the presence of only few transplanted cells caudally localized, as observed in KO brains (Fig. 6A). In possible good accordance with the limited grafting, NPC treatment recovered the memory defect that distinguishes Het mice from WT (Fig. 6D), but did not improve their general well-being (Fig. 6B) and motor skills (Fig. 6C).

## NPCs induce the deregulation of immune-related molecular pathways and the activation of IFNγ signaling

Bulk RNA-seq of WT and *Mecp2* KO cerebellum was performed to reveal the molecular mechanisms driven by the treatment. This cerebral region was selected considering the distribution of NPCs after transplantation and for its roles in motor and cognitive functions (Koziol et al, 2014) which were improved by NPCs. PCA analysis shows a clear separation between WT and KO cerebella, whereas NPC transplanted samples are more similar to their controls (Fig. 7A). Using a fold change >1 or <−1 as cut off for differential expression, we identified only a small percentage of DEGs (~20%) differentiating KO from WT samples, therefore confirming that the lack of Mecp2 subtly modifies gene transcription (Sanfeliu et al, 2019) (Fig. 7B). Using the same filters and

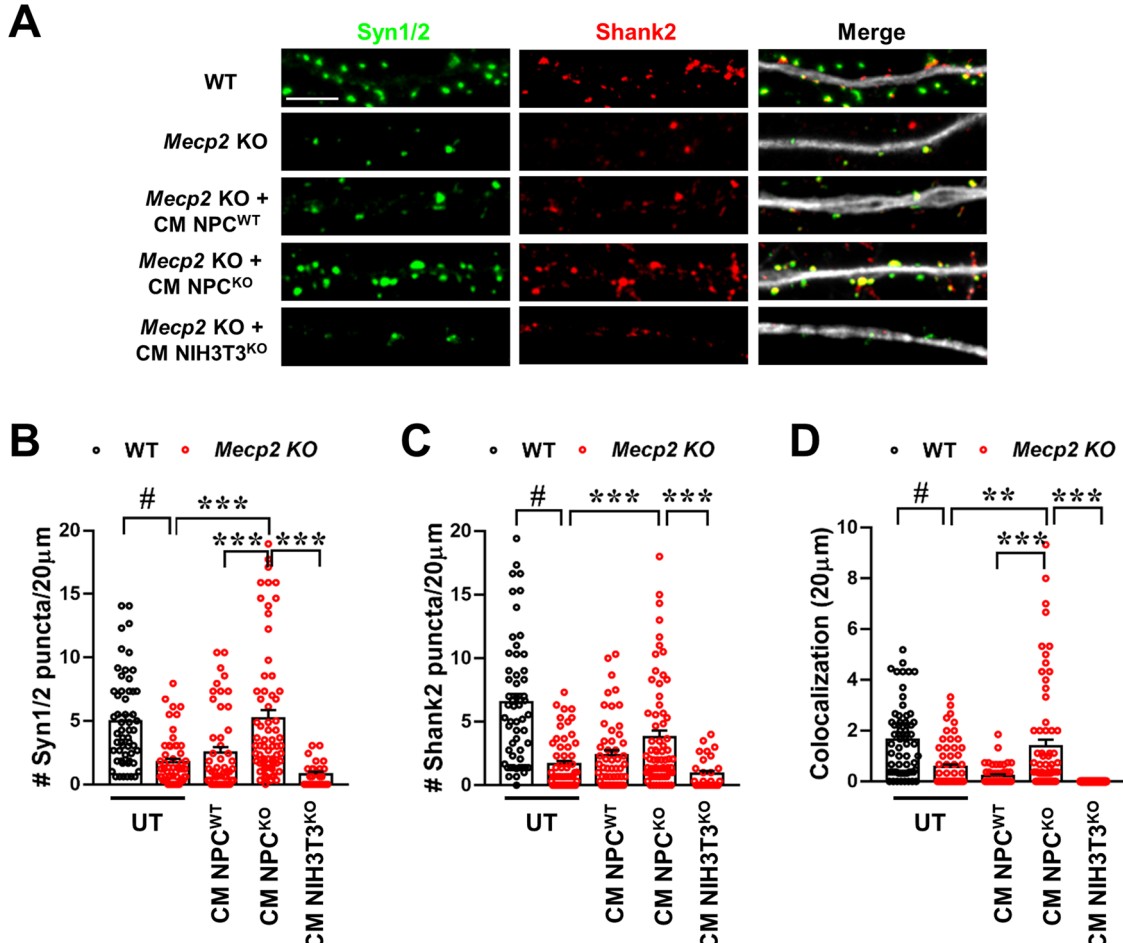

**Figure 3. NPCs ameliorate synaptic defects in *Mecp2* KO neurons by sensing the pathological environment.**

(A) Representative images of WT and KO neurons (DIV14) immunostained for Synapsin1/2 (Syn1/2; green), Shank2 (red), and Map2 (white) left untreated (UT) or treated with the conditioned medium (CM) collected from the co-cultures between WT neurons and NPCs (CM NPC^WT), or between KO neurons and NPCs (CM NPC^KO) or NIH3T3 (CM NIH3T3^KO). Scale bar = 5 μm. (B–D) Histograms show the mean ± SEM of Synapsin1/2 and Shank2 puncta density (B, C) and their colocalization (D). In (B): #p < 0.0001 WT *vs* KO by unpaired t test; ***p < 0.0001 KO *vs* KO + CM NPC^KO, ***p = 0.0003 KO + CM NPC^WT *vs* KO + CM NPC^KO, ***p < 0.0001 KO + CM NPC^KO *vs* KO + CM NIH3T3^KO by one-way ANOVA followed by Tukey post-hoc test. In (C): #p < 0.0001 WT *vs* KO by unpaired t test; ***p = 0.0006 KO *vs* KO + CM NPC^KO, ***p = 0.0001 KO + CM NPC^KO *vs* KO + CM NIH3T3^KO by one-way ANOVA followed by Tukey post-hoc test. In (D): #p < 0.0001 WT *vs* KO by unpaired t test; **p = 0.0023 KO *vs* KO + CM NPC^KO, ***p < 0.0001 KO + CM NPC^WT *vs* KO + CM NPC^KO, ***p < 0.0001 KO + CM NPC^KO *vs* KO + CM NIH3T3^KO by one-way ANOVA followed by Tukey post-hoc test. n = 56 KO, n = 51 KO + CM NPC^WT, n = 67 KO + CM NPC^KO, n = 28 KO + CM NIH3T3^KO. WT and KO neurons derived from a pool of 3 mice/genotype, whereas CM was collected from at least 6 different co-cultures per genotype and 3 independent experiments. Source data are available online for this figure.

comparing DEGs in transplanted KO tissues with respect to the untreated ones, we revealed a high number of DEGs; interestingly, genes with the higher entity of deregulation were upregulated (55 DEGs with a fold change >1). No transcriptional change was observed in NPC-injected WT samples, in good accordance with the lack of behavioral effects of NPCs in WT mice (Fig. 7B).

To identify the biological pathways enriched in the list of DEGs resulting from the different comparisons, we performed Over Representation Analysis (ORA). The ORA results of the comparison KO+PBS *versus* WT+PBS revealed the deregulation of several pathways related to RTT dysfunctions, such as synapse organization, ion channel transport and regulation of neurogenesis, thereby validating the RNA-seq analysis (Fig. EV3) (Gogliotti et al, 2018; Sanfeliu et al, 2019).

Interestingly, the results shown in (Fig. 7C), in which we analyzed the effects of NPCs in KO cerebella, revealed the deregulation of several pathways associated with immune response, including the pathway related to interferon gamma (IFNγ). By investigating the expression of genes specifically involved in this pathway, we revealed a general upregulation in NPC-treated KO *versus* KO, in contrast to a reduction in KO samples compared to WT (Fig. 7D). Coherently by gene set enrichment analysis (GSEA) (Subramanian et al, 2005) we found that IFNγ response was less represented in the KO cerebellum compared to WT (ES = −0.64; q-value = 0.001), whereas an enrichment was evident in KO+NPCs *versus* KO (ES = 0.80; q-value < 0.001) and WT (ES = 0.64; q-value < 0.001) (Fig. 7E). By qRT-PCR we analyzed the expression of a selected panel of genes associated with the IFNγ response, including Ppar9, Ifitm3, Irf8 and Bst2, validating the

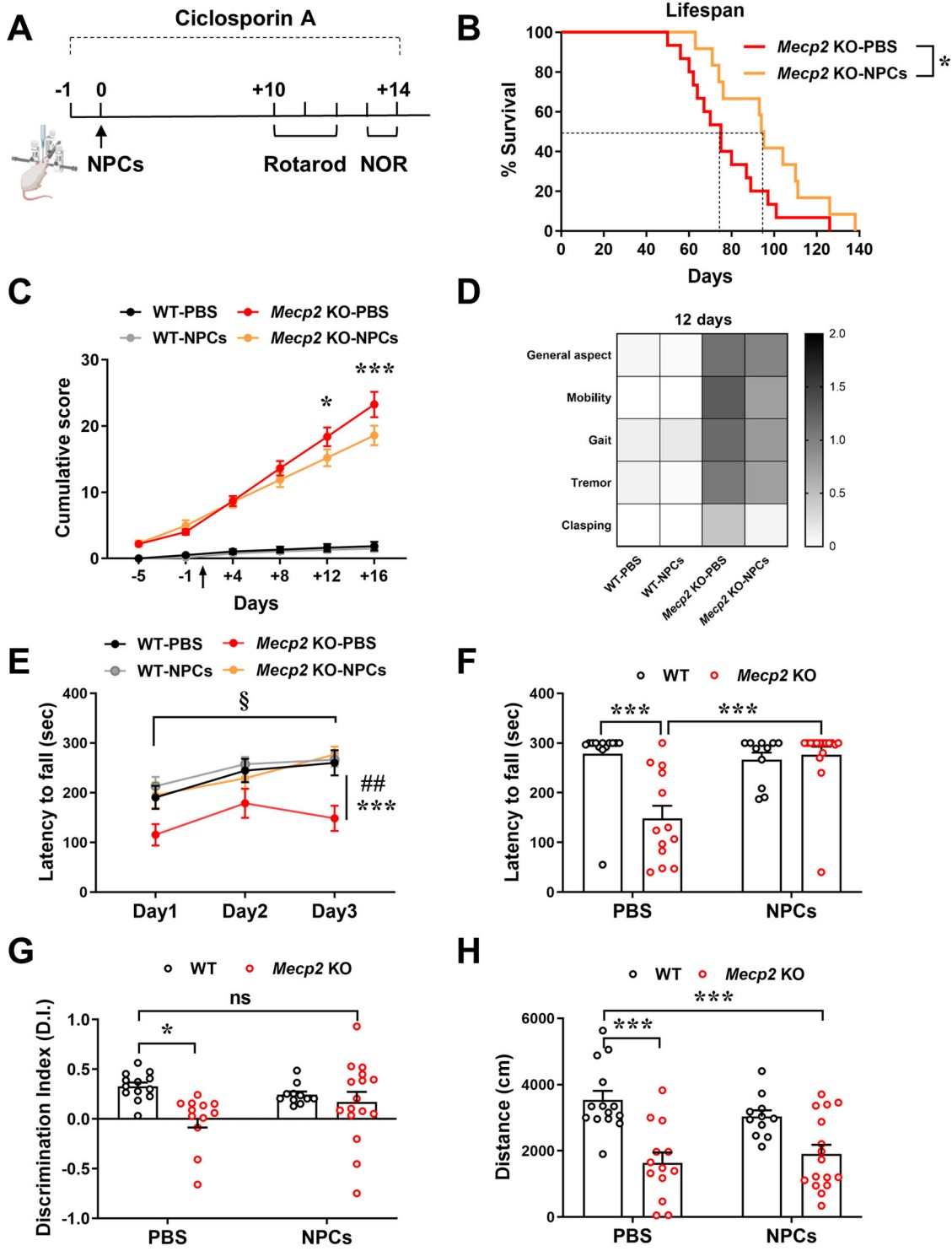

activation of this molecular pathway (Fig. 7F). To determine whether other brain regions had a similar transcriptional response to NPCs, RNA-seq was performed in the hippocampus, which is severely affected in RTT. The number of DEGs with p.adj < 0.05 in NPC-treated tissues *versus* untreated tissues was insufficient to perform bioinformatics analysis; we thus used a p.adj < 0.1. ORA analysis again suggested the involvement of the IFNγ pathway and the immune

system (Fig EV4A–C). Furthermore, when analyzing the genes involved in the IFNγ response pathway, we found that most of them were upregulated in the KO hippocampi after transplantation, whereas only a few were affected in the treated WT tissues, thereby reproducing the results in (Fig. 7D). Further, NPCs co-cultured with KO neurons (but not with WT) increased the levels of phosphorylated Stat1, that is considered a good marker of IFNγ pathway activation (Fig. 7G).

**Figure 4.  NPC transplantation prolongs the lifespan and ameliorates RTT-like impairments in *Mecp2* KO mice.**

(A) NPCs were injected in WT and KO mice (P45-47) and behavioral tests were conducted starting from 10 days after transplantation. Ciclosporin A (50 mg/kg, s.c.) was daily administered both in PBS- and NPC-treated WT/KO mice starting the day before surgery and for 15 days. (B) Kaplan–Meyer survival analysis shows that NPC transplantation prolongs the lifespan of KO mice compared to PBS-treated KO animals. The median survival corresponds to 70 days for PBS-KO mice and 94 days for NPC-KO mice. *$p = 0.0303$ by Gehan-Breslow-Wilcoxon test. $n = 13$ PBS-treated KO, $n = 11$ NPC-treated KO. (C) The graph depicts the mean ± SEM of the cumulative phenotypic score calculated for all the experimental groups. The black arrow indicates the day of transplantation. Asterisk denotes a significant difference between KO +NPCs and KO + PBS mice. The difference between KO + PBS and WT + PBS mice, or KO+NPCs and WT + PBS is omitted, although significant at all time-points, excluding at day −5. *$p = 0.0257$, ***$p = 0.0002$ by two-way ANOVA followed by Tukey post-hoc test. $n = 15$ WT + PBS, $n = 15$ WT+NPCs, $n = 8$ KO + PBS, $n = 12$ KO +NPCs. (D) Heatmap of the phenotypic score at the 12th day after NPC transplantation, indicating in a gray scale the severity for each symptom (from 0 = absent to 2 = very severe). $n = 15$ WT + PBS, $n = 15$ WT+NPCs, $n = 8$ KO + PBS, $n = 12$ KO+NPCs. (E) The graph represents the mean ± SEM of the time (in seconds) spent on the rod during each day of the test, thus deducing motor learning. Asterisks and hashtags denote a significant difference between KO+NPCs and KO + PBS mice, and between KO + PBS and WT + PBS mice, respectively, at day 3. § indicates a significant difference both in WT + PBS and in KO+NPCs in between the performance at day 3 and day 1. ##$p = 0.0033$, ***$p = 0.0003$, §$p = 0.0002$ in WT + PBS and §$p < 0.0001$ in KO+NPCs by two-way ANOVA followed by Tukey post-hoc test. (F) The histogram shows the mean ± SEM of the time (in seconds) spent on the rod at the 3rd day of the test. ***$p = 0.0001$ WT *vs* KO, ***$p < 0.0001$ KO *vs* KO+NPCs by two-way ANOVA followed by Tukey post-hoc test. (G) The histogram represents the mean ± SEM of the discrimination (D.I.) index values, defined as: time exploring the novel object – time exploring the familiar object)/total time) assessed by novel object recognition (NOR) test. *$p = 0.0162$ by two-way ANOVA followed by Tukey post-hoc test; ns indicated no statistically significant difference between NPC-treated KO and PBS-treated WT mice. (H) The histogram shows the distance (in cm) travelled during the first day of NOR test, expressed as mean ± SEM. ***$p < 0.0001$ WT *vs* KO, ***$p = 0.0003$ WT *vs* KO+NPCs by two-way ANOVA followed by Tukey post-hoc test. In (E-H), $n = 14$ WT + PBS, $n = 12$ WT+NPCs, $n = 11$ KO + PBS, $n = 16$ KO+NPCs. Source data are available online for this figure.

## The cytokine IFNγ holds therapeutic potential for the treatment of RTT

Instructed by molecular results, which pointed to the ability of NPCs to activate the IFNγ pathway in cellular and animal models of RTT, and encouraged by previous evidence indicating the neuroprotective effects of the cytokine (Filiano et al, 2016), we tested the potential benefits of an IFNγ administration.

We exposed cultured RTT neurons to IFNγ and investigated its ability to rescue synaptic defects. Three doses (25, 75, and 100 ng/ml) were initially supplemented to the medium, revealing that none of them was toxic (Fig. EV5A). WB analysis confirmed that all concentrations were able to induce Stat1 phosphorylation (Fig. EV5B). Notably, by using immunofluorescence, we proved that the cytokine reverted the density of synaptic markers and their colocalization in KO neurons in a dose-dependent manner (Fig. 8A–D). The highest dose of IFNγ significantly enhanced also the defective density of pre-synaptic puncta in Het neurons (Fig. 8E,F). No detrimental effect was observed in WT neurons (Fig. 8A–F).

Finally, we verified IFNγ efficacy in vivo. KO mice and WT littermates (P45) were i.c.m. injected with recombinant IFNγ (20 ng/ml) or an equal volume of vehicle. After 24 h, a cohort of mice was tested for memory functions, a second one for motor abilities (Fig. 9A). By NOR test, we reported that IFNγ significantly rescued memory deficits in KO mice (Fig. 9B). Similarly, by Rotarod test, we found that defects in coordination were recovered by the acute cytokine injection, without an improvement in motor learning (Fig. 9C,D). On the contrary, IFNγ was unable to ameliorate defects in the frequency of respiration and inspiration/expiration time in KO mice measured by whole-body plethysmography (Fig. 9E–H). Having observed a strong enhancement of memory in KO mice, we investigated the efficacy of an intracerebral injection of IFNγ on cognitive performance also in Het animals (P90). Importantly, the acute treatment with the cytokine rescued short-term memory also in Het mice (Fig. 9J).

## Discussion

Stem cell therapy represents a promising therapeutic option for the treatment of many neurodegenerative diseases (Lindvall and Kokaia, 2010). Although several stem cell types are available for therapeutic treatments, neural precursor cells (NPCs) offer the advantages of being highly stable in terms of self/renewal, expansion, and differentiation, and, importantly, do not originate tumors in vivo (Foroni et al, 2007). However, adult stem cells, including NPCs, can induce cancer when transplanted in a microenvironment different from their original one, thereby raising serious safety concerns (Melzi et al, 2010). These results reinforced the importance of treating neurological disorders with homotypic adult NPCs. Accordingly, several clinical trials based on NPC transplantation have been conducted so far (Genchi et al, 2023; Fan et al, 2023).

Although NPCs can differentiate into neurons, astrocytes, and oligodendrocytes, their benefic effects upon transplantation are not exclusively based on cell replacement. Indeed, transplanted NPCs often maintain an undifferentiated phenotype and, by sensing the encountered microenvironment, engage into the release of different therapeutic molecules, thus influencing many biological processes (Lindvall and Kokaia, 2010; Drago et al, 2013; Bacigaluppi et al, 2016; De Feo et al, 2017; Willis et al, 2020).

While literature supporting the effectiveness of stem cells, and in particular of NPCs, for neurodegenerative disorders is wide (Lindvall and Kokaia, 2010), only little evidence is available regarding their application for neurodevelopmental diseases (NDDs) (Siniscalco et al, 2018; Donegan and Lodge, 2020). Most of these studies investigated the therapeutic efficacy of mesenchymal or hematopoietic stem cells on animal models of autism spectrum disorders (ASD) and in a limited number of cases amniotic epithelial or adipose- and urine-derived stem cells were used (Nabetani et al, 2023). Concerning RTT, the ability of mesenchymal stem cells to increase neurogenesis and the number of synapses in the brain of *Mecp2* KO mice was recently reported; however, stem cells were engineered to constantly produce BDNF (Kim et al, 2021). Further indications of a positive effect of NPC transplantation for NDDs can be deduced from a Chinese publication in which human NPCs (hNPCs) derived from aborted fetal tissues were administered to 13 children with RTT and 9 with ASD (Liu et al, 2013).

This evidence, together with the known widespread neurobiological abnormalities of the RTT brain, which could induce NPCs

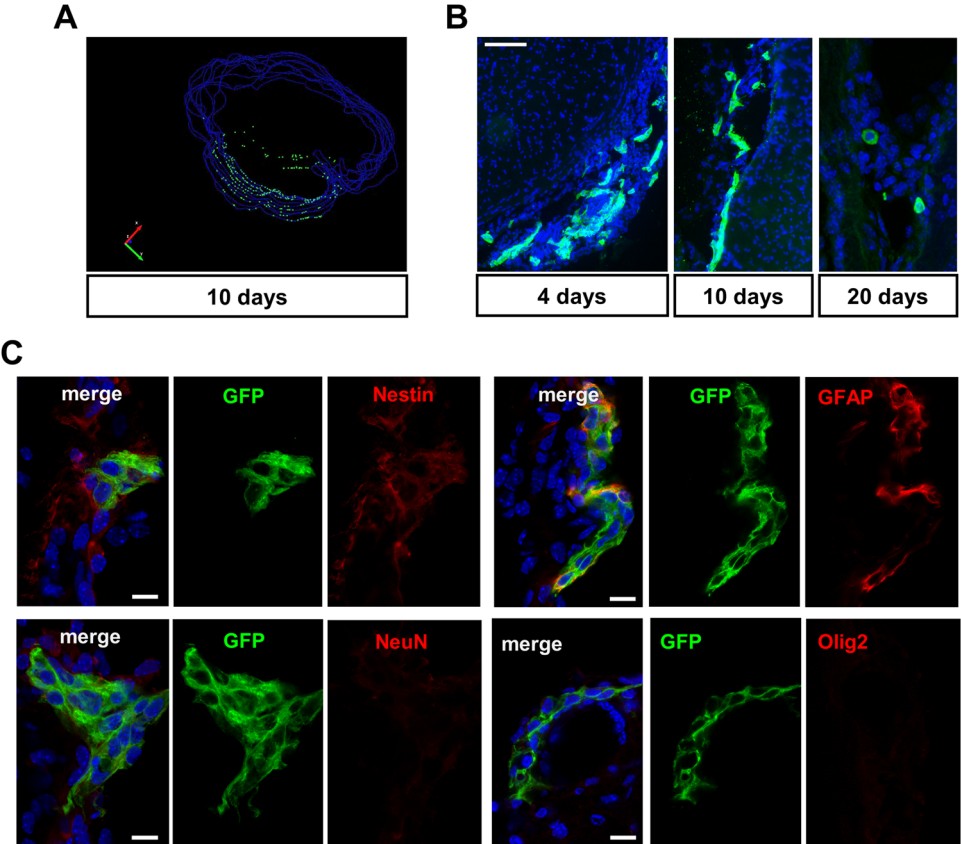

**Figure 5. NPCs localize along the meninges in *Mecp2* KO brain and mainly retain an undifferentiated phenotype.**

(A) 3D reconstruction of NPCs' distribution in the KO brain by Neurolucida software. (B) Representative images of GFP+-NPCs (green) localized along the meninges in the caudal region of the brain at 4, 10, and 20 days after transplantation ($n = 3$/time point). Nuclei are immunostained with DAPI (blue). Scale bar = 50 µm. (C) Representative images of GFP+-NPCs (green) with different cell markers (red). Scale bar = 20 µm. Source data are available online for this figure.

to secrete neuroprotective molecules, prompted us to study the therapeutic effect of NPCs in RTT mouse models. In vitro we proved the existence of a beneficial crosstalk between NPCs and RTT neurons, that is not established with WT cells, and that induces NPCs to release factors promoting synaptic rescue. This favorable crosstalk does not appear to be due to a different neuron/glia ratio because the same proportion is maintained under all culture conditions. These results led us to assess the value of NPC transplantation in the *Mecp2* deficient mice. Coherently with previous studies performed in mouse models of neurodegenerative disorders, only few grafted cells were retrieved in the transplanted brain and a progressive decline of their number was observed. The localization of grafted cells and their undifferentiated phenotype suggest that the positive effects should depend on a bystander mechanism. The therapeutic efficacy of NPCs emerged from the motor improvements and the partial amelioration of the cognitive functions. Lifespan in KO mice was also significantly prolonged; curiously, the observed delay in mortality correlates well with the duration of sustained NPC grafting. We hypothesize that a longer lasting and possibly a more robust benefit might result from repetitive transplantations.

The potential of NPCs was also tested in Het female mice, which closely mimic the genetic condition of RTT patients. Rescued memory functions were displayed by Het mice upon transplantation, whereas no improvement was observed in motor abilities. We believe that several factors contributed to the limited efficacy observed, including (i) a less graft-friendly environment, probably due to milder neurobiological alterations; (ii) the greater dispersion of results that typically derives from Het mice; (iii) the need to optimize the time window for treatment, as the disease progresses slower in Het females (Ribeiro and MacDonald, 2020).

To explore the mechanisms involved in the neuroprotective effects exerted by NPCs in RTT animals, we investigated the transcriptional changes induced by transplantation, highlighting the upregulation of numerous pathways associated with the immune response, among which we identified the pathway related to IFNγ as a strong candidate. Indeed, IFNγ is a soluble molecule, mainly secreted by immune cells, which can be easily targeted for therapeutic purposes (Monteiro et al, 2017). Further, this cytokine has been implicated in neuropsychiatric disorders (Arolt et al, 2000) and was already used in clinical settings as an immunomodulatory drug (Miller et al, 2009). Contrasting data are available regarding the effects of IFNγ on neurological symptoms, with some evidence reporting neurotoxic effects and other indicating neuroprotective ones (Mizuno et al, 2008; O'Donnell et al, 2015). This heterogeneity of data might depend on the source of the cytokine,

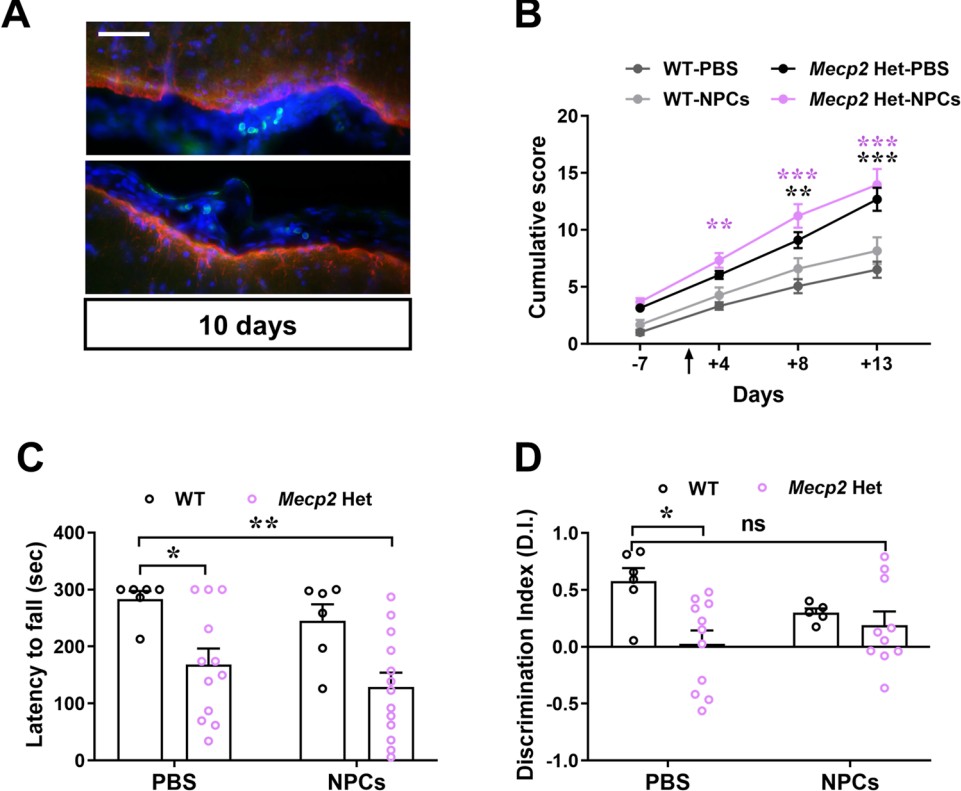

**Figure 6. NPC transplantation in *Mecp2* Het animals only improves their cognitive defects, without ameliorating their general well-being and motor abnormalities, in accordance with their limited grafting.**

(A) Representative image of GFP⁺-NPCs (in green) and GFAP (in red) in the caudal part of the Het brain 10 day after transplantation. Scale bar = 100 μm. (B) The graph depicts the mean ± SEM of the cumulative phenotypic score in female WT and Het mice injected with NPCs at P180. The black arrow indicates the day of transplantation. Black asterisk denotes a significant difference between Het+PBS and WT+PBS mice, whereas violet asterisk indicates a significant difference between Het+NPCs and WT+PBS mice. At day 4: **$p = 0.0022$; at day 8: **$p = 0.0020$, ***$p < 0.0001$; at day 13: ***$p < 0.0001$ by two-way ANOVA followed by Tukey post-hoc test. $n = 8$ WT + PBS, $n = 6$ WT+NPCs, $n = 11$ Het+PBS, $n = 11$ Het+NPCs. (C) The graph shows the mean ± SEM of the time (in seconds) spent on the rod during the 3ʳᵈ day of the test. *$p = 0.0474$, **$p = 0.0039$ by two-way ANOVA followed by Tukey post-hoc test. $n = 6$ WT + PBS; $n = 6$ WT+NPCs; $n = 12$ Het+PBS; $n = 13$ Het+NPCs. (D) The histogram represents the mean ± SEM of the discrimination (D.I.) index values, assessed by novel object recognition (NOR) test. *$p = 0.0175$ by two-way ANOVA followed by Tukey post-hoc test; ns indicated no significant difference between NPC-treated Het and PBS-treated WT mice. $n = 5$ WT + PBS; $n = 5$ WT+NPCs; $n = 11$ Het+PBS; $n = 10$ Het+NPCs. Source data are available online for this figure.

the activated pathway and the targeted pathology. Notably, the ability of IFNγ to rescue social behavior defects in immuno-suppressed mice was recently proved (Filiano et al, 2016).

A reduced concentration of IFNγ in the serum of RTT patients has been demonstrated (Leoncini et al, 2015), but single-cell sequencing analysis of meningeal immune cells reported increased IFNγ expression in KO mice (Li et al, 2023). Our data demonstrate that NPCs activate IFNγ response pathway in the KO cerebellum. However, they do not inform on the source of the cytokine, since both NPCs and resident immune cells might concur to the observed activation (Cossetti et al, 2014; Monteiro et al, 2017). The observation that NPC transplantation induces the upregulation of the same pathway also in the hippocampus, a brain region distal from grafted NPCs, suggests the involvement of resident cells. We also do not know which cells respond to the cytokine. However, the in vitro activation of Stat1 upon NPC exposure and the effectiveness of IFNγ on synaptic defects prompt us to indicate neurons as target cells also in vivo.

In accordance with its ability to ameliorate synaptic alterations in *Mecp2* deficient neurons, a single injection of IFNγ in the CSF of KO and Het animals completely reverted motor and memory impairments, which mainly rely on alterations of synaptic plasticity. In contrast, respiratory alterations were not improved by the treatment, suggesting that amelioration of respiratory defects might require the stabilization of less plastic molecular mechanisms.

In conclusion, we demonstrated that, by sensing the pathological milieu, NPCs secrete factors that rescue defects of RTT neurons and ameliorate neurological abnormalities of *Mecp2* deficient mice. Of relevance, from a therapeutic point of view, the efficacy of NPCs does not depend on the presence of functional MeCP2. Although the molecular mechanisms underpinning these beneficial effects might be different in vitro and in vivo, transcriptomic analysis strongly indicated the involvement of IFNγ. Indeed, treatment with IFNγ of RTT neurons and *Mecp2* deficient mice ameliorated morphological and behavioral defects, respectively. Although we are aware that many other factors might mediate the beneficial

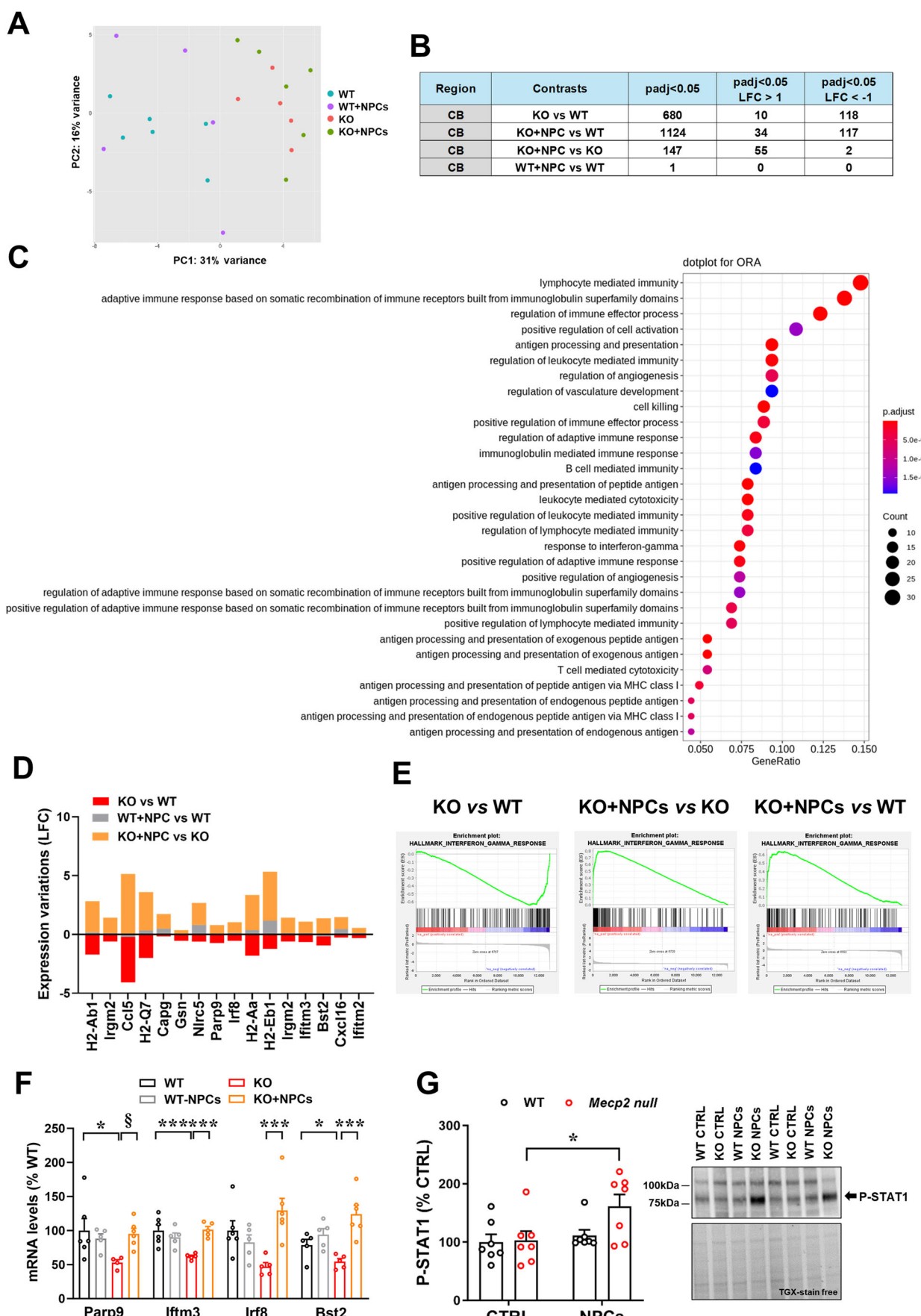

◄ **Figure 7. Bulk-RNA sequencing revealed the activation of IFNγ pathway in the transplanted *Mecp2* KO cerebellum.**

(A) Principal component analysis (PCA) plot for the sequenced samples of WT ($n = 6$), KO ($n = 5$), WT+NPCs ($n = 5$) and KO+NPCs ($n = 6$) cerebella. Percentage of variance is reported for both PC1 (first component) and PC2 (second component). (B) The number of deregulated genes (DEGs) for the different comparison is reported, considering a p.adj < 0.05. The number of DEGs with a LogFoldChange lower than −1 (down-regulated genes) or LogFoldChange greater than 1 (upregulated genes) is also indicated. (C) Dot plot of Gene Ontology (GO) enriched pathway analysis in the cerebellum, indicating the top 30 most enriched pathways of the comparison between KO +NPCs *versus* KO samples. For a description of the statistical method implemented in DESeq2 or ORA see Boyle et al (2004) and Yu et al (2012), respectively. (D) The graph shows the LogFoldChange (LFC) of genes belonging to the GO pathway "Interferon-γ response" in the comparisons KO *versus* WT, WT + NPC *versus* WT, and KO +NPCs *versus* KO. (E) Gene set enrichment analysis (GSEA) of IFNγ response in cerebella, indicating a significant enrichment of the gene set in KO+NPCs vs KO comparison, as well as in KO+NPCs vs WT comparison. (F) The histogram reports the transcriptional levels of genes associated to IFNγ pathway. Data, expressed as percentage of WT, are shown as mean ± SEM. Parp9: *$p = 0.0292$ WT *vs* KO, §$p = 0.0508$ KO *vs* KO+NPCs; Iftm3: ***$p = 0.0007$ WT *vs* KO, ***$p = 0.0008$ KO *vs* KO +NPCs; Irf8: ***$p = 0.0002$ KO *vs* KO+NPCs; Bst2: *$p = 0.0274$ WT *vs* KO, ***$p = 0.0010$ KO *vs* KO+NPCs by two-way ANOVA followed by Sidak's post-hoc test. $n = 5$ WT, $n = 6$ WT+NPCs, $n = 5$ KO, $n = 6$ KO+NPCs. (G) Western blot analysis of phosphorylated STAT1 (P-STAT1) in WT or KO neurons cultured with NPCs (for 14 days) ($n = 7$ mouse embryos/genotype). Data are represented as mean ± SEM and expressed as percentage of WT neurons cultured alone. *$p = 0.0242$ by two-way ANOVA followed by Sidak's post hoc test. Representative bands of P-STAT1 and the corresponding lanes of TGX-stain free gel are reported. Source data are available online for this figure.

effects of NPCs on RTT models, we believe that a pre-clinical study assessing the therapeutic efficacy of a prolonged treatment of IFNγ in RTT deserves further investigations.

# Methods

## Methods and protocols

### Animals

The experiments were performed on the *Mecp2*^tm1.1Bird mouse model in the outbred CD1 genetic background, generated by crossing *Mecp2* heterozygous (Het) females in C57BL/6 background (B6.129P2(C)-Mecp2^tm1.1Bird/J) with CD1 wild-type (WT) male

### Reagents and tools table

| Reagent/Resource | Reference or Source | Identifier or Catalog Number |
|---|---|---|
| Earl's Balanced Salt Solution (EBSS) | Sigma-Aldrich | Cat#E2888 |
| L-Cysteine | Sigma-Aldrich | Cat#C7352 |
| EDTA | Sigma-Aldrich | Cat#E6511 |
| Papain | Sigma-Aldrich | Cat#P4762 |
| Neurocult Basal Medium | Stem Cell Technologies | Cat#05702 |
| Proliferation Supplement | Stem Cell Technologies | Cat#05701 |
| FGF | Provitro | Cat#1370950500 |
| EGF | Provitro | Cat#1325051000 |
| Heparyn | Sigma-Aldrich | Cat#H3393-100KU |
| Accumax | Sigma-Aldrich | Cat#A7089 |
| Hank's Buffered Salt Solution (HBSS) | Thermo Fisher Scientific | Cat#14175-095 |
| Trypsin | Euroclone | Cat#ECB3052D |
| 0.25% trypsin/EDTA | Thermo Fisher Scientific | Cat#25200-056 |
| Fetal Bovine Serum (FBS) | Thermo Fisher Scientific | Cat#10500064 |
| Dulbecco's Modified Eagle Medium (DMEM) | Thermo Fisher Scientific | Cat#41966-029 |
| L-Glutamine | Sigma-Aldrich | Cat#G7513 |
| Penicillin/Streptomycin | Sigma-Aldrich | Cat#P0781 |
| Neurobasal | Thermo Fisher Scientific | Cat#21103049 |
| B27 Supplement | Thermo Fisher Scientific | Cat#A3582801 |
| Poly-D-lysine hydrobromide | Sigma-Aldrich | Cat#P7886 |
| MTT formazan | Sigma-Aldrich | Cat#M2003 |
| D-PBS | Euroclone | Cat#ECB4004L |
| Normal Goat Serum (NGS) | Thermo Fisher Scientific | Cat#50197Z |
| Fluoromount mounting medium | Sigma-Aldrich | Cat#F4680 |
| PureZOL RNA isolation Reagent | Bio-Rad | Cat# 7326890 |

| Reagent/Resource | Reference or Source | Identifier or Catalog Number |
|---|---|---|
| Glycogen RNA grade | Thermo Fisher Scientific | Cat# R0551 |
| DNase I Amplification Grade | Sigma-Aldrich | Cat# AMPD1 |
| SYBR Green Master Mix | Applied Biosystems | Cat# 4472908 |
| Bovine Serum Albumin (BSA) | Sigma-Aldrich | Cat# A3059 |
| Triton X-100 | Sigma-Aldrich | Cat# T8787 |
| DAPI | Thermo Fisher Scientific | Cat# 62248 |
| Precision Plus Protein All Blue | Bio-Rad | Cat# 1610373 |
| Tween® 20 | Sigma-Aldrich | Cat# P1379 |
| WESTAR SUN ECL | CYANAGEN | Cat# XLS063 |
| WESTAR ANTARES ECL | CYANAGEN | Cat# XLS0142 |
| 4x Laemmli Sample Buffer | Bio-Rad | Cat# #1610747 |
| 2-Mercaptoethanol | Sigma-Aldrich | Cat# M6250 |
| Xtra Taq Pol RTL GL | GENESPIN | Cat# XSTS-T5XRTL GL |
| Xtra RTL GL Reaction Buffer 5X | GENESPIN | Cat# XSTS-T5XRTL GL |
| Deoxynucleotide Set, 100 mM | Sigma-Aldrich | Cat# DNTP100A |
| **Experimental models** | | |
| Mecp2[tm1.1Bird] CD1 mouse model (M.Musculus) | Laboratory of Nicoletta Landsberger at University of Milan | (Cobolli Gigli et al, 2016) |
| C57BL/6 (M.Musculus) | Charles River Laboratories | |
| NIH3T3 fibroblasts | ATCC | RRID:CVCL0594 |
| **Recombinant DNA** | | |
| pCAG vector with an iresGFP empty | Addgene | |
| **Antibodies** | | |
| Mouse monoclonal anti-GFP | Thermo Fisher Scientific | Cat#A10262; RRID:AB_2534023 |
| Mouse monoclonal anti-GFAP (clone GA5) | Millipore | Cat#MAB 3402; RRID:AB_94844 |
| Rabbit polyclonal anti-Olig2 purified IgG | Abcam | Cat#ab136253; RRID:AB_2810961 |
| Mouse monoclonal anti-NeuN (clone 60) | Millipore | Cat#MAB377; RRID:AB_2298772 |
| Mouse monoclonal anti-Nestin (clone rat-401) | Immunological Science | Cat#MAB353; RRID:AB_94911 |
| Rabbit monoclonal anti-Map1 (D5G1) XP® antibody | Cell Signalling | Cat#8707; RRID:AB_2722660 |
| Chicken polyclonal anti-Synapsin1/2 IgY fraction | Synaptic System | Cat#106006; RRID:AB_2622240 |
| Mouse monoclonal anti-Shank2 purified IgG | Synaptic System | Cat#162211; RRID:AB_2661874 |
| Rabbit polyclonal anti-phosphorylated Tyr701 Stat1 (clone 58D6) | Cell Signalling | Cat#9167; RRID:AB_561284 |
| Rabbit polyclonal anti-Bdnf | Abcam | Cat# ab108319; RRID:AB_10862052 |
| **Oligonucleotides and other sequence-based reagents** | | |
| Forward primer for null allele: 5'-ACCTAGCCTGCCTGTACTTT-3' | Metabion | N/A |
| Forward primer for wt allele: 5'-GACTGAAGTTACAGATGGTTGTG-3' | Metabion | N/A |
| Reverse primer: 5'-CCACCCTCCAGTTTGGTTTA-3' | Metabion | N/A |
| List of primers for qRT-PCR | Metabion | Table S1 |
| **Chemicals, enzymes and other reagents** | | |
| Recombinant IFNγ | Immunological Sciences | GRF-15448 |
| Ciclosporin A | Novartis | |
| Tetrodotoxin citrate (TTX) | Tocris | Cat#1069 |

| Reagent/Resource | Reference or Source | Identifier or Catalog Number |
|---|---|---|
| **Software** | | |
| Fiji/ImageJ | Fiji | https://imagej.nih.gov/ij/ RRID:SCR_002285 |
| Neurolucida | MBF Bioscience | http://www.mbfbioscience.com/neurolucida RRID:SCR_001775 |
| NIS-Elements | Nikon | https://www.microscope.healthcare.nikon.com/products/software/nis-elements RRID:SCR_014329 |
| pClamp-10 software | Molecular Devices | http://www.moleculardevices.com/products/software/pclamp.html RRID:SCR_011323 |
| GraphPad Prism 8 | GraphPad Software LLC | https://www.graphpad.com RRID:SCR_002798 |
| RStudio | RStudio | RRID:SCR_000432 |
| Gene Set Enrichment Analysis | UC San Diego and Broad Institute | https://www.gsea-msigdb.org/gsea/index.jsp RRID:SCR_003199 |
| Uvitec Nine Alliance Software | Uvitec Cambridge | https://www.uvitec.co.uk/ |
| QuantStudio 5 Data Analysis Software | Thermo Fisher Scientific | https://www.thermofisher.com/us/en/home/global/forms/life-science/quantstudio-3-5-software.html |
| BioRender | BioRender | https://biorender.com |
| **Other** | | |
| PDL coated coverslips | Neuvitro | Cat#GG-12-PDL |
| Thincert™ Cell Culture Inserts 6 Well plates, tc, transparent membrane (PET), pore diameter 0.4 μm | Greiner Bio-One | Cat#GR657641 |
| Thincert™ Cell Culture Inserts 24 Well plates, tc, transparent membrane (PET), pore diameter 0.4 μm | Greiner Bio-One | Cat#GR662641 |
| 4–15% Criterion™ TGX Stain-Free™ Protein gel, 26 well, 15 μl | Bio-Rad | Cat#5678085 |
| Trans-Blot® Turbo™ Midi Nitrocellulose Transfer Packs | Bio-Rad | Cat#1704159 |
| Phire animal tissue direct PCR kit | Thermo Fisher Scientific | Cat#F140WH |
| Agilent RNA 6000 Nano Kit | Agilent Technologies | Cat#5067-1511 |
| TruSeq® Stranded mRNA Library Prep | Illumina | Cat#20020594 |
| RT² First Strand Kit | Qiagen | Cat#33404 |

mouse and maintaining animals on a clean CD1 background. CD1 *Mecp2* mutant line recapitulates the typical phenotype of *Mecp2* mutant animals in C57BL/6 background, with the advantage of higher breeding success and larger litters therefore facilitating basic and translational studies (Cobolli Gigli et al, 2016). Neural Precursor Cells were prepared from adult female C57BL/6 WT animals and adult male CD1 *Mecp2* KO mice. Both CD1 WT male mice and C57BL/6 WT females were purchased from Charles River Laboratories. Mouse genotype was determined by PCR using the following primers: 5′-ACCTAGCCTGCCTGTACTTT-3′ forward primer for null allele; 5′-GACTGAAGTTACAGATGGTTGTG-3′ forward primer for wild type allele; 5′-CCACCCTC-CAGTTTGGTTTA-3′ as common reverse primer.

For neuronal cultures, WT and *Mecp2* mutant embryos were generated by mating Het females with WT mice. The day of vaginal plug was considered E0.5 and primary neurons were prepared from E15.5 embryos. For in vivo and ex vivo experiments, *Mecp2* KO animals (P45/47) and *Mecp2* Het mice (P90 and P180), and the corresponding WT littermates, were randomly assigned to the treatment groups. Animals were sacrificed by rapid decapitation or by transcardiac perfusion depending on experimental needs. For

behavioral experiments, animals from the same litters were randomly allocated to the experimental groups.

Animals were housed in a temperature- and humidity-controlled environment in a 12 h light/12 h dark cycle with food and water *ad libitum*. All procedures were performed in accordance with the European Union Communities Council Directive (2010/63/EU) and Italian laws (D.L.26/2014). Protocols were approved by the Italian Council on Animal Care in accordance with the Italian law (Italian Government decree No. 175/2015-PR, No. 210/2017 and No. 187/2022-PR).

## Cell cultures

### Neural precursor cells (NPCs)

Adult female C57BL/6 WT mice (6 to 8 weeks old, 18–20 g) and male CD1 *Mecp2* KO animals (6 weeks old) were anaesthetized by an intraperitoneal injection of Tribromoethanol (250 mg/kg, i.p.) and the brain was removed and positioned in sterile PBS. Brain coronal sections were taken 3 mm from the anterior pole of the brain, excluding the optic tracts and 3 mm posterior to the previous cut. The subventricular zone (SVZ) of the lateral ventricles was

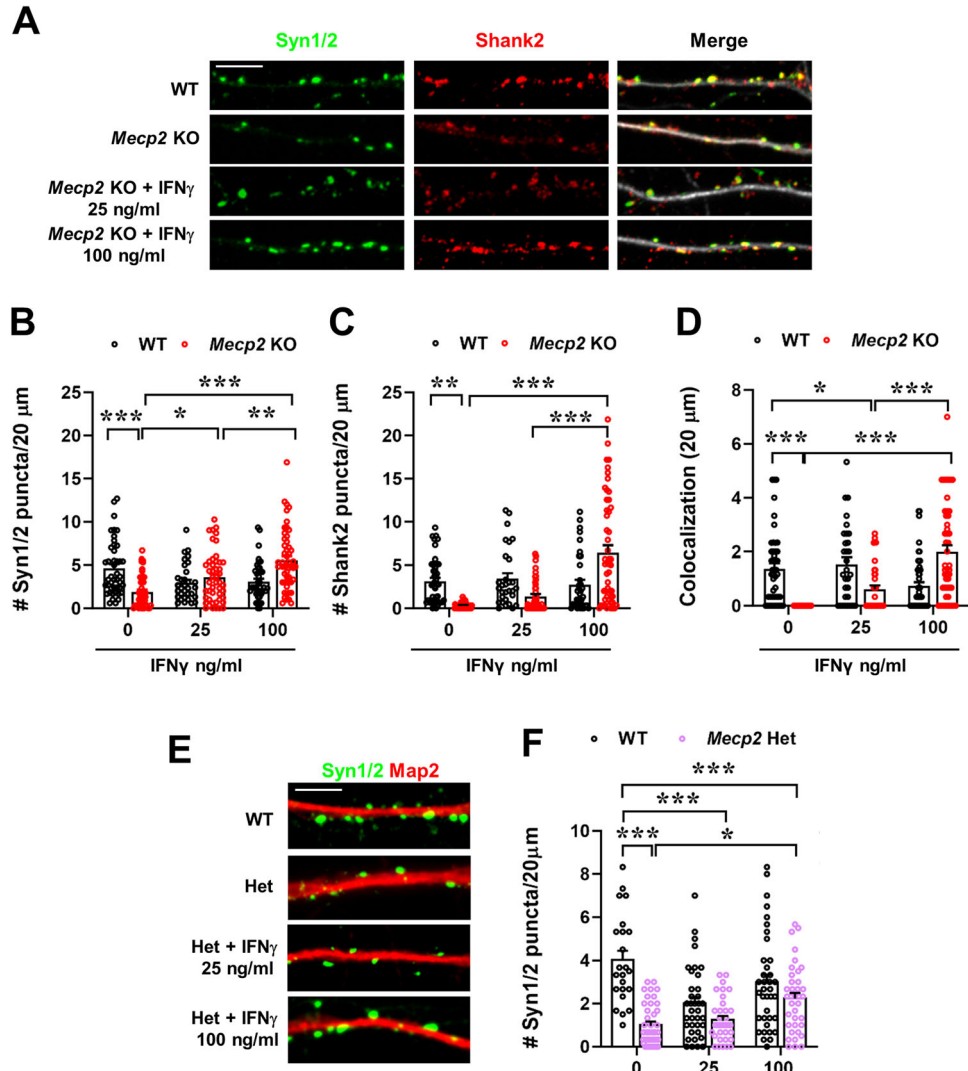

**Figure 8. Synaptic defects of *Mecp2* KO cortical neurons are improved by IFNγ treatment.**

(A) Representative images of WT and KO neurons (DIV14) untreated or treated for 24 h at DIV13 with IFNγ (25 and 100 ng/ml) and immunostained for Synapsin1/2 (Syn1/2; green), Shank2 (red) and Map2 (white). Scale bar = 5 µm. (B–D) Histograms report the mean ± SEM of the number of puncta counted in 20 µm for Synapsin1/2 (B), Shank2 (C), and colocalized puncta (D). In (B): ***p < 0.0001 WT *vs* KO, *p = 0.0379 KO *vs* KO + IFNγ 25 ng/µl, **p = 0.0047 KO + IFNγ 25 ng/µl *vs* KO + IFNγ 100 ng/µl, ***p < 0.0001 KO *vs* KO + IFNγ 100 ng/µl by two-way ANOVA followed by Tukey post-hoc test. In (C): **p = 0.0050 WT *vs* KO, ***p < 0.0001 KO + IFNγ 25 ng/µl *vs* KO + IFNγ 100 ng/µl, ***p < 0.0001 KO *vs* KO + IFNγ 100 ng/µl by two-way ANOVA followed by Tukey post-hoc test. In (D): ***p < 0.0001 WT *vs* KO, ***p < 0.0001 KO *vs* KO + IFNγ 100 ng/µl, ***p < 0.0001 KO + IFNγ 25 ng/µl *vs* KO + IFNγ 100 ng/µl, *p = 0.0447 WT vs KO 25 ng/µl by two-way ANOVA followed by Tukey post-hoc test. n = 44 WT, n = 32 WT + IFNγ 25 ng/ml, n = 40 WT + IFNγ 100 ng/ml, n = 47 KO, n = 47 KO + IFNγ 25 ng/ml, n = 52 KO + IFNγ 100 ng/ml. Neurons derived from at least 3 mice/genotype, from 2 independent experiments. (E) Representative images of WT and Het neurons (DIV14) untreated or treated for 24 h at DIV13 with IFNγ (25 and 100 ng/ml) and immunostained for Synapsin1/2 (Syn1/2; green) and Map2 (red). Scale bar = 5 µm. (F) The graph shows the mean ± SEM of Synapsin1/2 puncta density. *p = 0.0118 Het *vs* Het+IFNγ 100 ng/µl, ***p < 0.0001 WT *vs* Het, ***p < 0.0001 WT vs Het+IFNγ 25 ng/µl, ***p = 0.0006 WT vs Het+IFNγ 100 ng/µl by two-way ANOVA followed by Tukey post-hoc test. n = 23 WT, n = 37 WT + IFNγ 25 ng/ml, n = 37 WT + IFNγ 100 ng/ml, n = 49 Het, n = 35 Het+IFNγ 25 ng/ml, n = 37 Het+IFNγ 100 ng/ml. Neurons derived from at least 3 mice/genotype, from 2 independent experiments. Source data are available online for this figure.

isolated from the coronal section using iridectomy scissors. Tissues derived from at least two mice were pooled and digested for 30 min at 37 °C with Earl's Balanced Salt Solution (EBSS) (#E2888, Sigma-Aldrich) containing 200 mg/l L-Cysteine (#C7352, Sigma-Aldrich), 200 mg/l EDTA (#E6511, Sigma-Aldrich), 2 U/ml Papain (#P4762, Sigma-Aldrich) (Pluchino et al, 2003). Sample was centrifuged at 200 × g for 12 min, the supernatant was removed and the pellet was mechanically disaggregated with 2 ml of EBSS. The pellet was centrifuged again at 200 × g for 12 min and then dissociated with a

pipette. Cells were plated in Neurocult proliferation medium, containing Neurocult Basal Medium (#05702, Stem Cell Technologies), Proliferation Supplement (#05701, Stem Cell Technologies), 10 ng/ml FGF (#1370 9505 00, Provitro), 20 ng/ml EGF (#1325 0510 00, Provitro), 0.0002% Heparyn (#H3393-100KU, Sigma-Aldrich) and 1% Penicillin/Streptomycin (P/S; #P0781, Sigma-Aldrich). After approximately one week, a small percentage of the isolated cells begun to proliferate, giving rise to neurospheres, which grow in suspension. Neurospheres were centrifuged at 50 × g

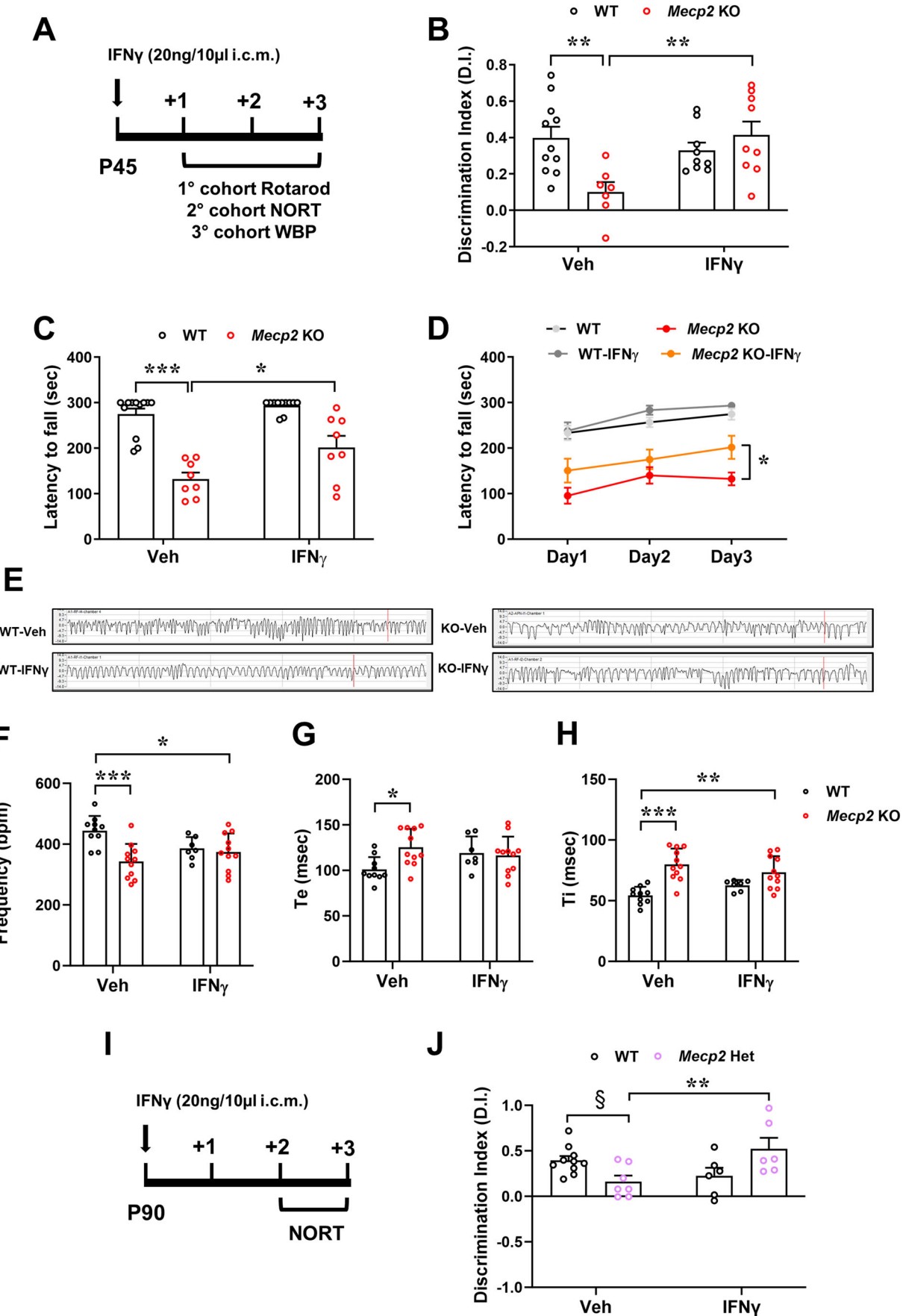

**Figure 9. IFNγ injection in *Mecp2* deficient mice rescues motor and cognitive impairments.**

(A) Scheme of in vivo experiments in which IFNγ (20 ng/10 µl) or vehicle (Veh) was stereotaxically injected in WT and KO mice (P45; i.c.m.). Three cohorts of mice were used for assessing cognitive, motor and breathing defects. (B) The histogram shows the mean ± SEM of the discrimination (D.I.) index values. **$p = 0.0092$ WT *vs* KO, **$p = 0.0084$ KO *vs* KO + IFNγ by two-way ANOVA followed by Tukey post-hoc test. $n = 11$ WT+Veh, $n = 7$ KO+Veh, $n = 9$ WT + IFNγ, $n = 9$ KO + IFNγ. (C, D) The graphs represent the mean ± SEM of the time (in seconds) spent on the rod during each day of the test reflecting motor learning abilities. In (C), the graph shows the mean ± SEM of the time (in seconds) spent on the rod at the 3rd day of the test. ***$p < 0.0001$ and *$p = 0.0178$, by two-way ANOVA followed by Tukey post-hoc test. In (D), the graph reports the time spent on the rod during each day of the Rotarod test. *$p = 0.0342$ by two-way ANOVA followed by Tukey post-hoc test. $n = 11$ WT+Veh, $n = 8$ KO+Veh, $n = 10$ WT + IFNγ, $n = 8$ KO + IFNγ. (E) Representative traces from the WBP analysis of WT and KO mice treated with vehicle or IFNγ. (F–H) The histograms show the mean ± SEM of the frequency, time of expiration (Te) and inspiration (Ti) measured by WBP in WT and KO mice 72 h after injection. In (F), *$p = 0.0234$ and ***$p = 0.0006$; in (G), *$p = 0.024$; in (H), **$p = 0.0014$ and ***$p < 0.0001$ by two-way ANOVA followed by Tukey post-hoc test. $n = 10$ WT+Veh, $n = 11$ KO+Veh, $n = 7$ WT + IFNγ, $n = 11$ KO + IFNγ. (I) Schematic representation of the in vivo experiments, in which IFNγ (20 ng/10 µl) or vehicle (Veh) was stereotaxically injected in WT and Het mice (P90; i.c.m.) before testing short-term memory functions. (J) The graph represents the mean ± SEM of the discrimination (D.I.) index values assessed by NOR test. §$p = 0.0593$ and **$p = 0.0080$ by two-way ANOVA followed by Sidak's post-hoc test. $n = 10$ WT+Veh, $n = 7$ Het+Veh, $n = 6$ WT + IFNγ, $n = 6$ Het+IFNγ. Source data are available online for this figure.

for 10 min, the supernatant removed, 200 ml of Accumax (#A7089, Sigma-Aldrich) was added and the tube was incubated at 37 °C for 10 min. The pellet was mechanically dissociated and cells were plated at a density 7500 cells/cm² for passages or used for in vitro and in vivo experiments.

For in vivo experiments, NPCs were infected with $3 \times 10^6$ T.U./ml of a third-generation lentiviral vectors. Cells were dissociated 12 h before infection and plated at high density ($1.5 \times 10^6$ cells in a 75 cm² flask) in 10 ml of medium. 48 h after infection, cells were harvested, centrifuged at $200 \times g$ for 12 min and plated without dissociation at a 1:1 ratio. After 3 passages in vitro, FACS analysis was performed to verify the efficiency of the infection (Pluchino et al, 2003). All cells used were negative for mycoplasma.

### Primary cortical neurons

At E15.5, WT, KO, and Het mouse embryos were used to prepare neuronal cultures (Frasca et al, 2020). Brains were removed under a microscope and immersed in ice-cold Hank's Buffered Salt Solution (HBSS; #14175-095, Thermo Fisher Scientific). Meninges were gently removed, cerebral cortex was rapidly dissected and maintained in ice-cold HBSS. Tissues were incubated with 0.25% trypsin/EDTA (# 25200-056, Life Technologies) for 7 min at 37 °C and the digestion was blocked with 10% FBS (#10500064, Thermo Fisher Scientific) in DMEM (#41966-029, Life Technologies). Cortices were then mechanically dissociated in DMEM, containing 10% FBS, 1% L-glutamine (#G7513, Sigma-Aldrich), 1% P/S. Neurons were seeded in neuronal medium [Neurobasal (#21103049, Thermo Fisher Scientific), 2% B27 (#A3582801, Thermo Fisher Scientific), 1% L-Glutamine, 1% P/S] on coated glass coverslip (Neuvitro) in 24-well plates (40,000 cells/well) for immunofluorescence analysis, in poly-D-lysine (0.1 mg/ml; #P7886, Sigma-Aldrich)-coated 6-well plates (200,000 cells/well) for western blot experiments and in 96-well plates (10,000 cells/well) for MTT assay (#M2003, Sigma-Aldrich). In these experimental conditions, astrocytes corresponded to ~10% of total cells, oligodendrocytes to ~1%, while no microglial cell was detected, without a genotype effect (Fig EV1E).

To facilitate neuronal morphological analysis, by in utero electroporation we transfected some cortical neurons in embryos with a GFP expressing plasmid. Timed-gestation Het females (E13.5) were deeply anesthetized with Tribromoethanol (250 mg/kg; i.p.) and uterine horns were exposed by midline laparotomy. Plasmid DNA (pCAG vector with an iresGFP empty; 0.5 µg/µl of DNA) was injected in the ventricle using a pulled micropipette.

Then platinum electrodes were placed outside the uterus over the telencephalon and 4 squared pulses of 40 V were applied at 50 ms intervals. The uterus was then placed back in the abdomen; muscle and skin were closed with sutures.

### NIH3T3 fibroblast cultures

NIH3T3 fibroblasts were maintained in growth medium (DMEM supplemented with 10% FBS, 1% L-glutamine, 1% P/S) in T75 flasks at 37 °C with 5% $CO_2$. When confluent, cells were washed with PBS (#ECB4004L, Euroclone) and then incubated with trypsin (#ECB3052D, Euroclone) for 3–5 min at 37 °C. Trypsin was inactivated with growth medium, cells were then centrifuged at $1200 \times g$ for 7 min, and the pellet was mechanically disaggregated with 1 ml of medium. Cells used were tested and were negative for mycoplasma.

### NPCs/NIH3T3-neuron co-cultures

NPCs and NIH3T3 fibroblasts were seeded in neuronal medium on transwell inserts (pore size = 0.4 µm; Corning Costar), at a density 1:100 respect to neurons. When neurons attached to coverslips or wells (~2 h after plating), inserts with NPCs and NIH3T3 cells were carefully transferred on neurons and maintained until DIV7 or DIV14, without changing the medium.

From co-cultures between neurons and NPCs/NIH3T3, conditioned medium (CM) was collected at DIV14. CM was immediately centrifuged at 1200 rpm for 5 min at 4 °C to remove cellular debris and stored at −80 °C until use.

### Neuronal treatment

For the treatment of neurons with CM, the day of the experiment, CM samples were left at room temperature (RT) for 10 min, then heated at 37 °C in a water bath for 5 min and finally added to neurons, in a ratio 1:1 respect to neuronal culture medium. CM treatment was conducted for 24 h (DIV13-DIV14). To treat neurons with IFNγ, WT and *Mecp2* deficient neurons were treated with 25, 75, or 100 ng/ml of mouse recombinant IFNγ (GRF-15448; Immunological Sciences) or nuclease-free-$H_2O$ (vehicle) as control for 24 h (DIV13-DIV14). IFNγ was dissolved in sterile $H_2O$, subdivided in 5 µl-aliquots and stored at −20 °C.

### Intra-cisterna magna injection

Mice were anesthetized with a mixture of $O_2$ and 3% isoflurane and during surgery anaesthesia was maintained at 1.5% isofluorane

and respiration continuously monitored. Mice were fixed on a stereotactic device (David Kopf Instruments). Vehicle (10 μl of sterile PBS) or NPCs ($10^6$ cells/10 μl) were injected intra-cisterna magna (i.c.m.) in WT, *Mecp2* KO and Het mice. IFNγ administration was performed in WT, KO and Het animals. A 10 μl Hamilton syringe was placed between the atlas and occipital bone at 35°, and advanced to puncture the cisterna magna. The following stereotactic coordinates were used: $x = 0$; $y = -7.5$ mm from λ; $z = -4.5$ mm from muscle plane. A total volume of 10 μl was injected in a 5-min time period and the needle was placed in situ for 3 min after injection before being slowly removed. Since NPCs were injected in CD1 animals (allogenic transplant), mice were treated with subcutaneous Ciclosporin A (15 mg/kg, s.c; Novartis), starting the day before transplantation and for the consecutive 15 days. PBS-injected mice were similarly treated with the immunosuppressive drug.

## Behavioral analyses

All the behavioral experiments were performed by a researcher blind to the treatment and genotype.

### Phenotypic characterization

*Mecp2* KO, Het mice and WT littermates were tested for the presence or absence of RTT-like symptoms with a previously described scoring system (Guy et al, 2007; Cobolli Gigli et al, 2016; Gandaglia et al, 2019; Scaramuzza et al, 2021). At each session, an observer assigned a score for general condition, mobility, gait, hindlimb clasping and tremor. When the sum of the individual scores was >8, the mouse was euthanized for ethical reasons. Day of euthanasia was considered day of the death, without distinguishing it from natural death. Graphically, for each parameter the evolution of symptomatology was represented by a cumulative plot, obtained by plotting for each day the score summed to all preceding ones.

### Rotarod test

Rotarod test was set in a three-day paradigm using a 5 lane Ugo Basile's Rotarod for mice. Every day, each mouse was subjected to three trials. For each trial, animals were placed on the rotating rod (4 rpm) and were allowed to habituate for 10 s. Then the rotation speed was gradually accelerated every 30 s across a period of 300 s, from 4 rpm to 40 rpm. For each mouse, trial terminated when animal fell of the rod or achieved the maximum time (300 s). Rotarod test was performed 10 days after NPCs/vehicle injection, and 24 h after IFNγ/vehicle injection.

### Novel object recognition test

The novel object recognition (NOR) test was performed in a square arena of 45 × 45 cm (Balducci et al, 2017; Scaramuzza et al, 2021). On day 1, mice were first allowed to habituate to the testing arena in a 10 min session and mobility in the open field was monitored. On day 2, animals underwent the training phase (5 min), in which two identical objects were introduced into the arena, allowing the mouse to explore them. Finally, after 1 h from the training phase, mice were tested for their memory (5 min). The discrimination index (D.I.), defined as the difference between the exploration time for the novel object and the familiar object, divided by total exploration time, was calculated. The sessions were recorded with the video tracking software EthoVision XT 14 (Noldus). NOR test

was performed 13 days after NPCs/vehicle injection, and 48 h after IFNγ/vehicle injection.

### Plethysmography

The breathing activity of freely-moving mice was recorded in a Vivoflow whole-body plethysmography system (EMKA Technologies) using a constant flow pump connected to the animal chamber, thus ensuring proper inflow of fresh air. Four plethysmography chambers of 200 ml, calibrated by injecting 1 ml of air, were used for simultaneous measurements. Breathing cycles were recorded for 15 min under normal-ventilation after an adaptation phase of 10 min in the recording chamber during the previous 5 days. The signal was amplified and recorded with IOX2 software.

## Immunofluorescence

For immunofluorescence analysis on brain sections, animals were anesthetized with Tribromoethanol (250 mg/ml, i.p.) and transcardially perfused with 30 ml ice-cold PBS, followed by 50 ml 4% paraformaldehyde (PFA). Brains were removed and post-fixed in 4% PFA for additional 24 h at 4 °C, then cryoprotected for 48 h at 4 °C in 30% sucrose in PBS and frozen in n-pentane at −30 °C for 3 min. Coronal sections were cut with a cryostat (Leica Biosystems) along the rostro-caudal orientation of the brain and mounted on SuperFrost Plus microscope slides (Menzel-Glaser) or, alternatively, put on a multiwell in PBS supplemented with $NaN_3$.

Immunofluorescence on serial brain sections mounted on slides (20 μm in thickness) was performed to study GFP-positive cells distribution; free-floating immunofluorescence was conducted on slices (30 μm in thickness) to study GFP-positive cells differentiation. Brain sections were washed three times for 5 min in PBS and incubated for 1 h in blocking solution [10% normal goat serum (NGS; #50197Z, Thermo Fisher Scientific), 0.1% Triton X-100 in PBS]. Sections were then incubated overnight at 4 °C with the primary antibody diluted in 1% NGS, 0.1% Triton X-100 in PBS. The following primary antibodies were used: anti-GFP (1:500; #A10262, Thermo Fisher Scientific), anti-GFAP (clone GA5, 1:1000; #MAB 3402, Millipore), anti-Olig2 (1:200; #ab136253, Abcam), anti-NeuN (clone 60, 1:100; #MAB377, Millipore) and anti-Nestin (clone rat-401, 1:100; #MAB353, Immunological Science). Slices were washed 3 times for 10 min in PBS and then incubated for 1 h with the Alexa-Fluor secondary antibody (1:500 in blocking solution). Sections were washed 8–10 times in PBS for 5 min and incubated with DAPI (0.1 mg/ml in PBS; #62248, Thermo Fisher Scientific) to stain nuclei; sections were washed in PBS and finally mounted with the Fluoromount mounting medium (#F4680; Sigma-Aldrich). Images were acquired at Nikon Ti2 Microscope equipped with an A1+ laser scanning confocal system and a 100× oil-immersion objective. To estimate the number of transplanted NPCs in KO brains, GFP$^+$ cells were manually counted on serial brain sections under an epifluorescence microscope (Nikon Eclipse Ti). The number of cells per section was multiplied by the number of sections, thus obtaining overall number of GFP$^+$ NPCs per brain (Bacigaluppi et al, 2016).

For immunofluorescence on cultured cells, neurons (DIV7 or DIV14) seeded on glass coverslips were fixed for 8 min with 4% PFA dissolved in PBS with 10% sucrose, then washed 3 times with PBS and stored in 0.1% $NaN_3$ in PBS at 4 °C. Cells were permeabilized in 0.2% Triton X-100 in PBS for 3 min on ice. Cells

were washed in 0.2% BSA in PBS, then in blocking solution (4% BSA in PBS) for 15 min and finally incubated with primary antibodies overnight at 4 °C. Primary antibodies were diluted in 0.2% BSA in PBS as follow: anti-Map2 (clone D5G1, 1:1000; #8707, Cell Signalling), anti-Synapsin1/2 (1:500; #106006, Synaptic System), anti-Shank2 (1:300; #162211, Synaptic System). After washing in BSA 0.2% in PBS, cells were incubated with the specific Alexa Fluor secondary antibody (1:500 in BSA 0.2% in PBS) for 1 h at RT. After 5 washes in PBS, nuclei were stained with DAPI and cells were washed in PBS. Glass coverslips were mounted on microscope slides with Fluoromount mounting medium and stored at 4 °C until image acquisition.

## Analysis of neuronal morphology and synaptic markers

Dendritic branching and length were acquired in DIV7 neurons. GFP$^+$ neurons were acquired at epi-fluorescence microscopes (Nikon Eclipse Ti) using an excitation wavelength of 488 nm with a 20× objective. To characterize neuronal morphology, images were processed using ImageJ software. Sholl analysis plugin was used to study the complexity of the dendritic arbor and NeuronJ plugin to measure dendritic length (Patnaik et al, 2020). Since both plugins require binary masks, we manually traced all the GFP$^+$ dendrites with Photoshop for each neuron and obtained images were binarized. We analyzed dendritic arborization by performing Sholl analysis using a radius step size of 10 μm. The same reconstructed neurons were used to analyze the total lengths of dendrites by using NeuronJ.

To analyze synaptic puncta density and colocalization, we processed DIV14 neurons. Z-stacks images (127.28 × 127.28 μm$^2$, 1024 × 1024-pixel resolution, 16-bit greyscale depth) were acquired at Nikon Ti2 Microscope equipped with an A1+ laser scanning confocal system and a SR Apo TIRF 100× oil-immersion objective, using a step size of 0.3 μm. For each dataset, images were acquired in four channels (laser wavelength for DAPI: 409.1 nm; laser wavelength for Synapsin1/2: 487.5 nm; laser wavelength for Shank2: 560.5 nm; laser wavelength for Map2: 635.5 nm) and parameters were maintained constant within the experiment (offset background, digital gain, laser intensity, pinhole size, scanning speed, digital zoom, scan direction, line average mode).

Puncta density was calculated by counting synaptic puncta within a manually selected ROI (length 20 μm on 3 primary branches/neuron) by using ImageJ software. Only puncta with a minimum size of 0.16 μm$^2$ were counted using *Analyze Particles*. To assess puncta colocalization of pre- and post-synaptic markers, the plugin *Colocalization highlighter* was run on each Z-stack image. Colocalized puncta were quantified in manually selected ROIs of the binary mask created from the maximum intensity projection. Only puncta with a minimum size of 0.1 μm$^2$ were counted (Frasca et al, 2020; Albizzati et al, 2024).

## Electrophysiological measurements

Whole-cell patch-clamp recordings were obtained from WT and *Mecp2* Het cortical neurons at DIV14 in the voltage-clamp modality using the Axopatch 200B amplifier and the pClamp-10 software (Molecular Devices). Recordings were performed in Krebs'-Ringer's-HEPES (KRH) external solution (NaCl 125 mM, KCl 5 mM, MgSO$_4$ 1.2 mM, KH$_2$PO$_4$ 1.2 mM, CaCl$_2$ 2 mM, glucose

6 mM, HEPES-NaOH pH 7.4 25 mM). Recording pipettes were fabricated from glass capillary (World Precision Instrument) using a two-stage puller (Narishige); they were filled with the intracellular solution potassium-gluconate (KGluc 130 mM, KCl 10 mM, EGTA 1 mM, HEPES 10 mM, MgCl$_2$ 2 mM, MgATP 4 mM, GTP 0.3 mM) and the tip resistance was 3–5 MΩ. In order to identify excitatory postsynaptic currents in miniature (mEPSCs), cortical neurons were held at −70 mV and Tetrodotoxin (TTX) 1 μM was added to the external solution. The recorded traces have been analyzed using Clapfit-pClamp 10 software, after choosing an appropriate threshold.

## Western blot

Neurons at DIV14 were washed briefly with sterile PBS and collected in 15% 2-mercaptoethanol in sample buffer. Samples were heated at 95 °C for 5 min and then separated on a SDS-PAGE on TGX-stain free precast gel (4–15% of acrylamide gradient) and transferred on a nitrocellulose filter using a semi-dry transfer apparatus (TransBlot SD; Bio-Rad). Membranes were incubated for 1 h in blocking solution [5% BSA in 0.1% Tween-20 in Tris-buffered saline containing (TBST)], and incubated overnight at 4 °C with the following primary antibodies: rabbit anti-phosphorylated Tyr701 Stat1 (clone 58D6, 1:1000; #9167, Cell Signalling; RRID:AB_561284) or rabbit anti-Bdnf (1:1000; ab108319, Abcam; RRID:AB_10862052). After 3 washes in TBST, membranes were incubated with HRP-conjugated secondary antibody for 1 h at RT (1:10,000; Jackson ImmunoResearch). The immunocomplexes were visualized by using the ECL substrate (Cyanagen) and Essential V6 imaging platform, UVITEC system (Cleaver Scientific Ltd). Band density measurements were performed using UVITEC software. Results were normalized to total protein content visualized by a TGX stain-free method (Bio-Rad).

## RNA extraction, qRT-PCR, RNA sequencing, and bioinformatics

Twenty days after transplantation, WT and *Mecp2* KO animals were euthanised by cervical dislocation and brain quickly removed. Total RNA from cerebella was extracted using Purezol (Bio-Rad) and quantified using a NanoDrop spectrophotometer. RNA integrity was assessed by using *RNA 6000 Nano Reagent kit* on Agilent 2100 Bioanalyzer (Agilent Technologies) (Albizzati et al, 2022). All samples showed an RNA integrity number (RIN) > 7.5.

Bulk RNA-sequencing was conducted on WT ($n = 6$), *Mecp2* KO ($n = 5$), WT+NPCs ($n = 5$), *Mecp2* KO+NPCs ($n = 6$) cerebella and hippocampi, analyzed as individual samples. cDNA library of the collected RNA samples was obtained using the TruSeq RNA Library Prep kit from Illumina. Single-end RNA Sequencing was performed on an Illumina HiSeq 2500 Next Generation Sequencing instrument. The quality of the reads was verified with FastQC (v.0.11.8). Sequencing adapters were trimmed with Trimmomatic (v. 0.39) and the trimmed fastq files were aligned to the reference genome with STAR (v. 2.53a) (Dobin et al, 2013). The mouse reference genome used for the alignment was the Mus musculus GENCODE release M22 (GRCm38.p6). Finally, the mapped reads were counted, grouped by genes, with featureCounts (v. 1.6.4) setting the strandness of the single-end reads as 'reverse'. Quality check of the different steps of the analysis was performed with

**Table 1. List of primers used for qRT-PCR.**

| Gene | Primer FW | Primer REV |
|------|-----------|------------|
| Ifitm3 | TCTGCTGCCTGGGCTTCATAGC | GTAGGCCTGGGCTCCAGTCACA |
| Bst2 | ACCCAGGACAGTCTGCTGCAGG | TGCTGCTCCAGGGCTTGAGACA |
| Ifitm2 | TTCTTCAACGCCTGCTGCCTGG | GCACTTGGCAGTGGAGGCGTAG |
| Parp9 | CCGGGGCCTGACTCTCCAGATT | TCGACTGTGCCACTCGTCCTGA |
| Irf8 | CAAGAGGAGCCCATCCCCACCA | GGCATATCCGGTCACCAGTGGC |
| Hprt | ACAGGCCAGACTTTGTTGGAT | TGCAGATTCAACTTGCGCTC |
| Rpl13 | TGGCTGGCATCCACAAGAAA | TTCTTCAGCAGAACTGTCTCCC |

MultiQC (Ewels et al, 2016) (Appendix Fig. S1). Principal Component Analysis was performed using the prcomp function in R, using the 500 most variable genes in term of Reads Per Kilobase Million (RPKM).

Differential gene expression analysis was performed with the DESeq2 bioconductor package on the following comparisons: *Mecp2* KO *versus* WT, both treated with PBS as control (*Mecp2* KO + PBS, WT + PBS), to assess the effects of *Mecp2* deficiency; *Mecp2* KO treated with NPCs (*Mecp2* KO+NPCs) vs *Mecp2* KO + PBS, to assess the effects of the treatment on KO animals; WT treated with NPCs (WT+NPCs) vs WT + PBS, to assess the effects of the treatment on WT animals; *Mecp2* KO+NPCs vs W +-PBS, to assess the rescue on KO animals. A *p*-value adjusted with FDR < 0.05 was used to determine the significance of DEGs (Boyle et al, 2004).

Over Representation Analysis (ORA) was performed using the Bioconductor package clusterProfiler (Yu et al, 2012). The function 'simplify' was used to remove redundancy of enriched GO terms.

Gene Set Enrichment Analysis (GSEA) (Subramanian et al, 2005) (version 4.1.0) was performed using the R package fgsea (Korotkevich et al, 2021) on shrunken, log-normalized exonic fold changes from DESeq2. Background was set to all expressed genes in this study and 1000 permutations were set to generate a null distribution for enrichment score in the hallmark gene sets and functional annotation gene sets.

For qRT-PCR experiments, RNA was reversely transcribed using the RT$^2$ First Strand Kit (#330404, Qiagen) as instructed by the manufacturer. The resulting cDNA was used as a template with SYBR Green Master Mix (Applied Biosystems) with designated primers (Table 1). Melting curve showed a single product peak, indicating good product specificity. Hprt and Rpl13 were used as housekeeping genes and geometric average was calculated and used for the analysis of fold change in gene expression using the 2(−delta Ct) method.

### Statistical analysis

All data are expressed as mean ± SEM. Before any statistical analysis, normality distribution was assessed for each dataset by D'Agostino and Pearson tests. Outliers were evaluated by ROUT test (Q = 1%) or Grubb's test ($\alpha = 0.05\%$). No other exclusion criteria were applied. Sample size was calculated considering an error $\alpha = 0.05$, a power = 0.80 and an effect size that was determined on the basis of our previous data. Statistical significance for multiple group comparisons was determined by one- or two-way analysis of variance (ANOVA), followed by Tukey post-hoc

**The paper explained**

**Problem**

*MECP2* mutations cause RTT, the first genetic cause of severe intellectual disability in girls worldwide. Although potentially treatable, so far only a drug mainly affecting fine motor skills and communication is available. The identification of valid therapy is hampered by the still limited comprehension of the neurological roles of MeCP2 and the consequences of its loss of function. NPC transplantation has already been proved safe and efficacious in many neurological disorders but a comprehensive study of NPC efficacy in RTT is lacking.

**Results**

We have collected several data indicating that NPC transplantation improves RTT-like symptoms in *Mecp2* mutant animals and that NPC-secreted factors are responsible for the beneficial effects on *Mecp2* mutant neurons. Importantly, our study disclosed that the activation of the Interferon-γ pathway participates to the observed benefic effects; accordingly, we proved the therapeutic efficacy of this cytokine in RTT models.

**Impact**

Willing to respond to the unmet need of identifying novel therapies for RTT, we proved the therapeutic potential of NPCs and we identified Interferon-γ as a possible novel healing molecule for RTT. Although this study remains at the pre-clinical realm, the obtained positive results represent the "proof-of-principle" required to directly inform new clinical trials's design.

test. Mann–Whitney tests was used for comparing two groups. All statistical analyses were performed using GraphPad Prism 9.

## Data availability

RNA-seq data produced from this study have been deposited to the ArrayExpress database (https://www.ebi.ac.uk/biostudies/arrayexpress) and assigned to the identifiers (E-MTAB-12813 and E-MTAB-14403).

The source data of this paper are collected in the following database record: biostudies:S-SCDT-10_1038-S44321-024-00144-9.

## Peer review information

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

## Acknowledgements

This work was supported by the Italian parents' association "PRO RETT Ricerca" to AF and NL, by Banca d'Italia to AF, by Fondation Jérôme Lejeune to AF (JLF#1951), and by Ministero dell'Università e della Ricerca, PRIN – Research Project of National Relevance (2022TWKYL5) to NL. We are also grateful to "One day Sofia". We acknowledge NOLIMITS Unitech, an advanced

imaging facility at the University of Milan, and the Genomic Facility of the IRCCS San Raffaele Hospital (CTGB). We are very grateful to Dr. Alessandro Ianni, for his contribution to bioinformatics analysis and to Prof. Elena Borroni from University of Milan for her generous gift of antibody against phosphorylated Stat1. We are grateful to all members of NL and GM laboratories for helpful discussions. We finally acknowledge the Department of Medical Biotechnology and Translational Medicine and the SCALE-UP project.

## Author contributions

**Angelisa Frasca**: Conceptualization; Resources; Data curation; Formal analysis; Supervision; Funding acquisition; Investigation; Visualization; Methodology; Writing—original draft; Project administration; Writing—review and editing. **Federica Miramondi**: Formal analysis; Investigation; Visualization. **Erica Butti**: Resources; Investigation; Methodology. **Marzia Indrigo**: Investigation. **Maria Balbontin Arenas**: Formal analysis; Investigation; Visualization. **Francesca M Postogna**: Formal analysis; Investigation; Visualization. **Arianna Piffer**: Investigation. **Francesco Bedogni**: Investigation. **Lara Pizzamiglio**: Investigation. **Clara Cambria**: Investigation. **Ugo Borello**: Data curation; Validation. **Flavia Antonucci**: Formal analysis; Visualization. **Gianvito Martino**: Conceptualization; Resources. **Nicoletta Landsberger**: Conceptualization; Resources; Supervision; Funding acquisition; Writing—original draft; Project administration; Writing—review and editing.

Source data underlying figure panels in this paper may have individual authorship assigned. Where available, figure panel/source data authorship is listed in the following database record: biostudies:S-SCDT-10_1038-S44321-024-00144-9.

## Disclosure and competing interests statement

The authors declare no competing interests.

# Expanded View Figures

**Figure EV1. NPC effects on the morphological and functional properties of neurons.**

(A) Sholl analysis reports the capacity of NPCs to increase dendritic complexity also in WT neurons. The graph depicts the mean ± SEM of the number of intersections of WT neurons cultured alone, or with NIH3T3 or NPCs from DIV0 to DIV7. $*p = 0.0194$ at 30 μm, $*p = 0.0270$ at 50 μm, $*p = 0.0045$ at 60 μm, $*p = 0.0074$ at 70 μm, $*p = 0.0256$ at 80 μm, $*p = 0.0262$ at 130 μm, $*p = 0.0070$ at 140 μm by two-way ANOVA followed by Tukey post-hoc test. $n = 33$ WT, $n = 17$ WT + NIH3T3; $n = 39$ WT + NPC. (B) The histogram shows the mean ± SEM of the total number of intersections calculated by Sholl analysis for WT and KO neurons cultured alone, or in culture with NIH3T3 or NPCs from DIV0 to DIV7. $*p = 0.0135$ WT *vs* KO, $*p = 0.0275$ WT *vs* WT+NPCs, $**p = 0.0032$ KO *vs* KO+NPCs by two-way ANOVA followed by Tukey post-hoc test. $n = 33$ WT, $n = 17$ WT + NIH3T3; $n = 39$ WT + NPC; $n = 39$ KO; $n = 26$ KO + NIH3T3; $n = 32$ KO+NPCs. Neurons derived from at least 3 different mice/genotype. (C) Representative traces of excitatory postsynaptic current in miniature (mEPSCs) recorded in primary WT and Het neurons (left) and in Het neurons treated with NPC or NIH3T3 (right). (D) Histograms represent the mean ± SEM of the mEPSCs frequency (Hz) and amplitude (pA) both expressed as values normalized on the frequency of WT neurons or amplitude of WT neurons. $n = 26$ WT; $n = 24$ Het; $n = 19$ Het+NPCs; $n = 14$ Het+NIH3T3. Data derived from 3 independent experiments. $**p = 0.0025$ WT *vs* Het by Mann–Whitney test; $*p = 0.0405$ by one-way ANOVA followed by Dunn's test. (E) The graph reports the percentage of contamination ± SEM of astrocytes, oligodendrocytes and microglia in cortical neurons at DIV14, independently from the genotype. $n = 3$ for each genotype.

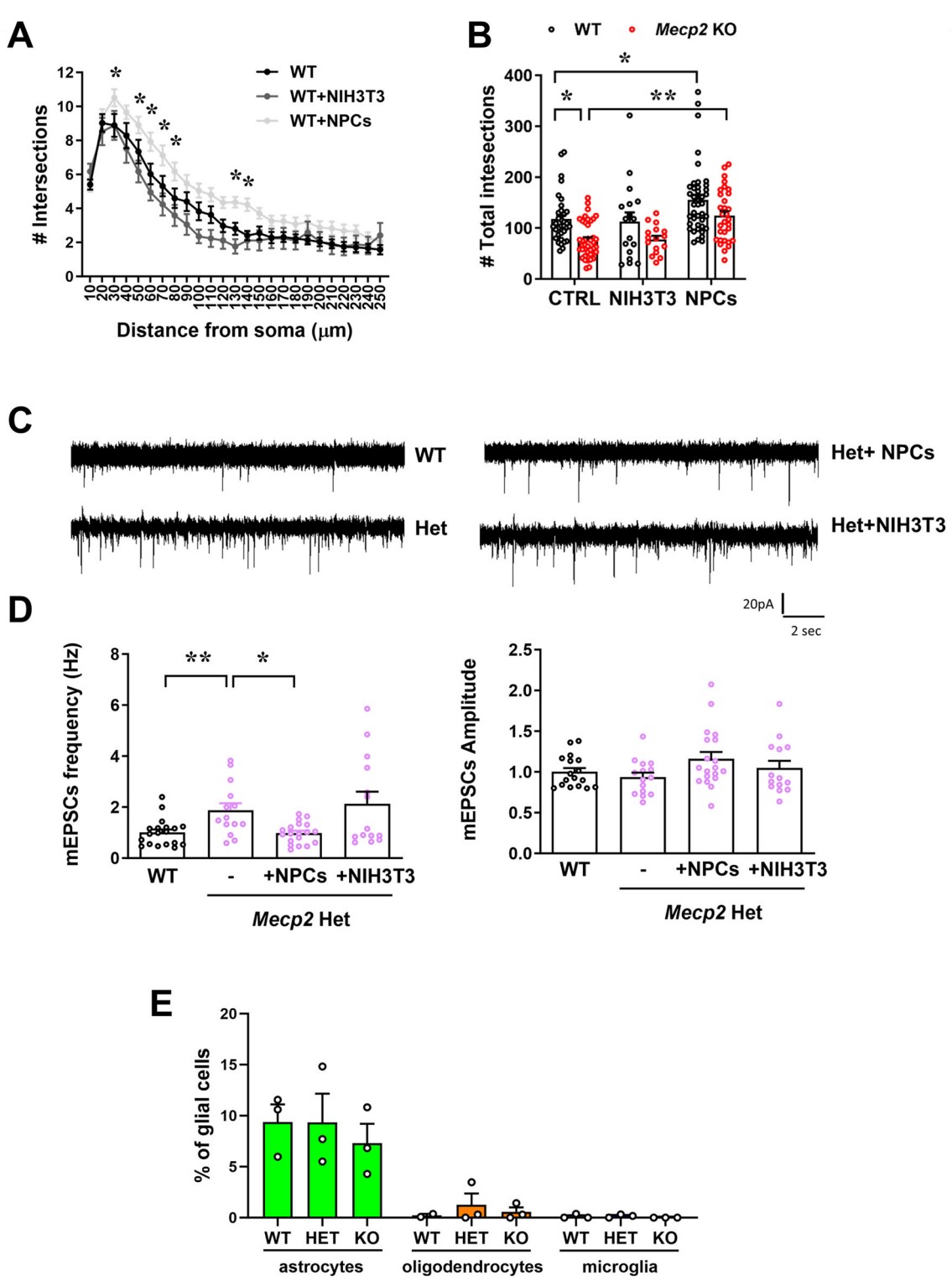

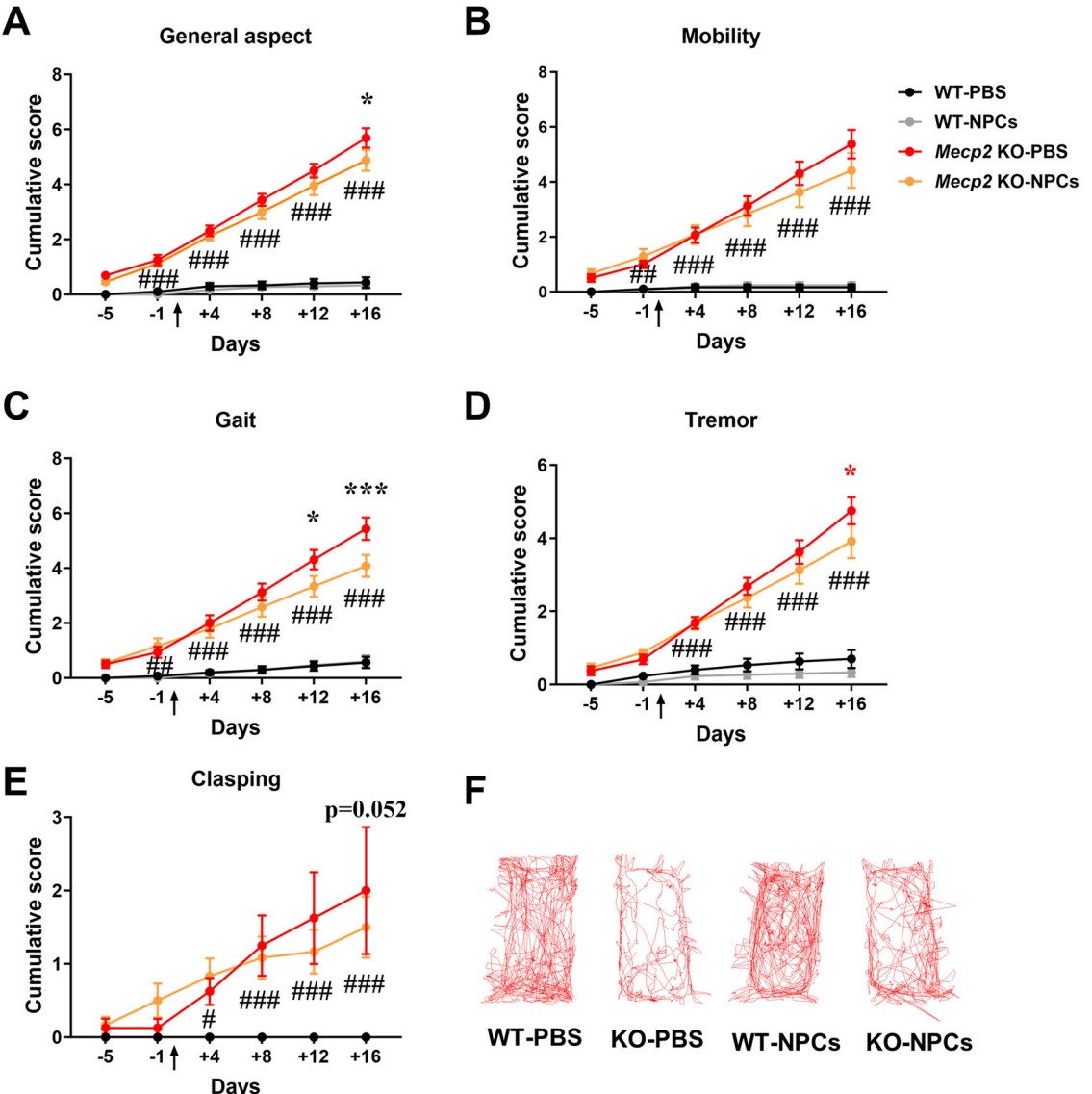

**Figure EV2. Phenotypic characterization of NPC-transplanted *Mecp2* KO mice.**

(A–E) Behavioral scoring was performed by a researcher blind to the treatment, assigning a score between 0 and 2 to general aspect (A), mobility (B), gait (C), tremor (D), and clasping (E). For each parameter, the graph reports its progression by a cumulative plot, in which the mean ± SEM of each value is obtained by summing the score of each day with those assigned the preceding days. Asterisks indicate a significant difference between KO + PBS and KO+NPCs; hashtags denote a difference between KO + NPC and WT mice. #$p = 0.0140$ for clasping, ##$p = 0.0010$ for gait, ##$p = 0.0022$ for mobility, ###$p < 0.0001$ for parameters, *$p = 0.0144$ for general aspect, *$p = 0.0224$ and ***$p = 0.0005$ for gait, *$p = 0.0367$ for tremor by two-way ANOVA followed by Tukey post-hoc test. $n = 15$ WT + PBS, $n = 15$ WT+NPCs, $n = 9$ KO + PBS, $n = 12$ KO+NPCs. (F) Representative traces of the distance travelled during the first day of NOR test by WT + PBS, KO + PBS, WT+NPCs and KO+NPCs mice. The corresponding data are reported in Fig. 4H.

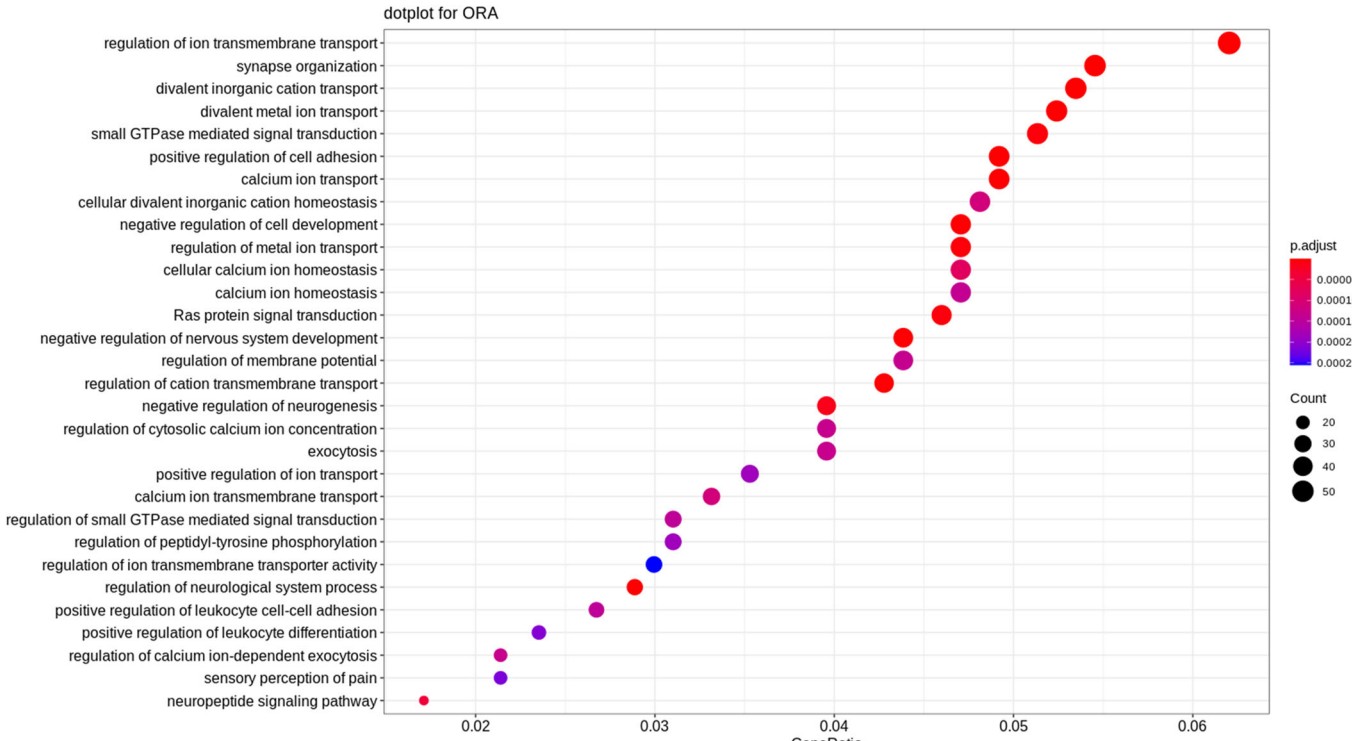

**Figure EV3.  Gene Ontology (GO) analysis.**

(**A**) Dot plot of Gene Ontology (GO) enriched pathway analysis in the cerebellum, indicating the top 30 most enriched pathways of the comparison between KO *versus* WT samples. For a description of the statistical method implemented in ORA see Yu et al, 2012.

**A**

| Region | Contrasts | padj<0.05 | padj<0.05 LFC > 1 | padj<0.05 LFC < -1 |
|--------|-----------|-----------|-------------------|--------------------|
| HP | KO vs WT | 1214 | 12 | 15 |
| HP | KO+NPC vs WT | 2074 | 54 | 26 |
| HP | KO+NPC vs KO | 9 | 4 | 0 |
| HP | WT+NPC vs WT | 5 | 0 | 0 |

**B**

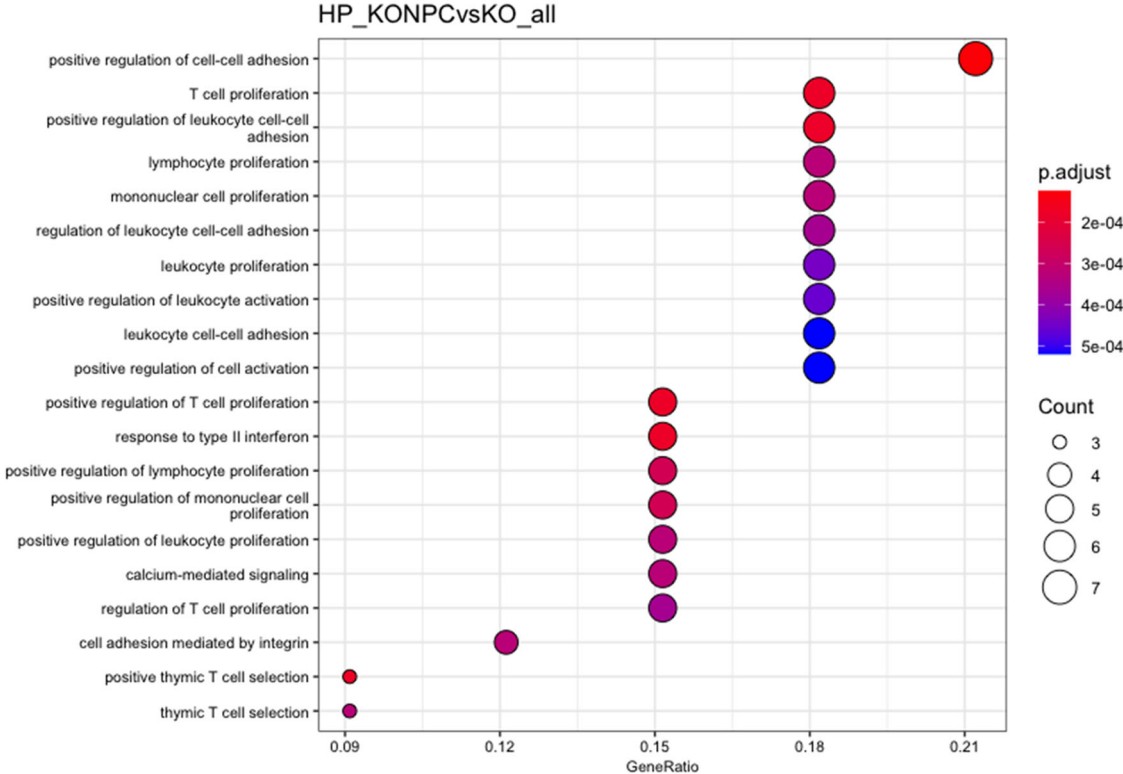

**C**

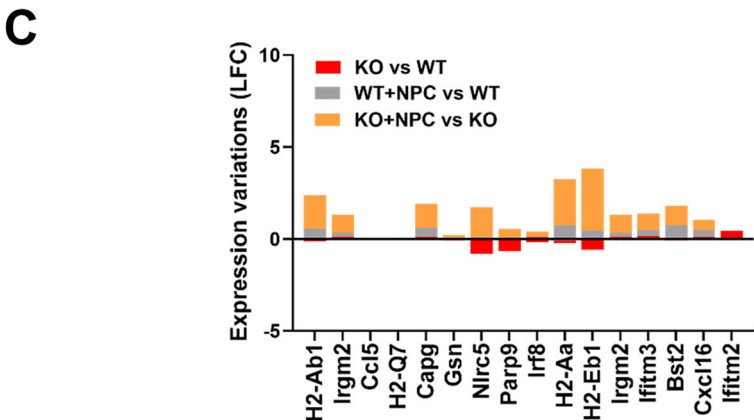

**Figure EV4.  Bulk RNA-sequencing indicated a mild upregulation of the IFNγ response in the hippocampus of KO mice after NPC transplantation.**

(A) The table reports the number of deregulated genes (DEGs) for the different comparisons, considering a p.adj < 0.05 and a p.adj < 0.1. The number of DEGs with a LogFoldChange lower than −1 (down-regulated genes) or LogFoldChange greater than 1 (upregulated genes) is also indicated. WT (n = 6), KO (n = 5), WT+NPCs (n = 5) and KO+NPCs (n = 6) hippocampi. (B) Dot plot of Gene Ontology (GO) indicating the top 20 most enriched pathways of the comparison between KO+NPCs *versus* KO samples. For a description of the statistical method implemented in DESeq2 or ORA see Boyle et al (2004) and Yu et al (2012), respectively. (C) The graph shows the LogFoldChange (LFC) of genes belonging to the GO pathway "Interferon-γ response" in the comparisons KO *versus* WT, WT + NPC *versus* WT, and KO+NPCs *versus* KO.

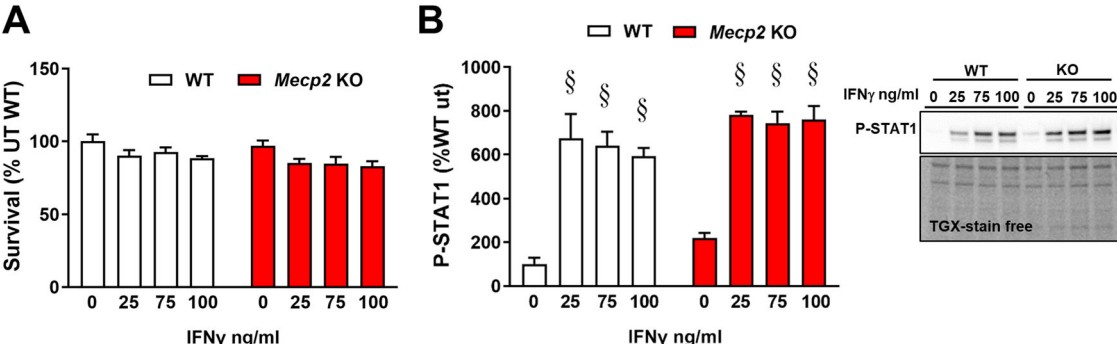

**Figure EV5. IFNγ does not affect neuronal survival and activates its downstream kinase in vitro.**

(A) The histogram represents the cell survival (% untreated WT neurons) ± SEM, assessed by MTT assay. IFNγ was added for 24 h in DIV13 primary neurons and three doses were tested: 25, 75, and 100 ng/ml. (B) The histogram reports the mean ± SEM of the levels of phosphorylated STAT1 after IFNγ treatment. Data are normalized to total protein content, visualized by a TGX stain-free technology. Representative bands of phosphorylated STAT1, and the corresponding lanes of TGX-stain-free gel, in WT and KO neurons treated or not with IFNγ are depicted. §$p = 0.0003$ WT-UT *vs* WT 100 ng/ml and §$p < 0.0001$ for the other comparisons, by two-way ANOVA followed by Tukey's post-hoc test. § denotes a significant difference respect to the corresponding untreated control of the same genotype. WT and KO neurons derived from 3 different mice/genotype.

