## [Peer Review File · EMBO Molecular Medicine]

Neural precursor cells rescue symptoms of Rett syndrome by activation of the Interferon γ pathway

Angelisa Frasca, Federica Miramondi, Erica Butti, Marzia Indrigo, Maria Balbontin, Francesca Postogna, Arianna Piffer, Francesco Bedogni, Lara Pizzamiglio, Clara Cambria, Ugo Borello, Flavia Antonucci, Gianvito Martino, and Nicoletta Landsberger

Corresponding author: Nicoletta Landsberger (nicoletta.landsberger@unimi.it)

Review Timeline:

Submission Date:	7th Jan 24
Editorial Decision:	1st Feb 24
Revision Received:	2nd Aug 24
Editorial Decision:	14th Aug 24
Revision Received:	29th Aug 24
Accepted:	4th Sep 24

Editor: Lise Roth

Transaction Report:

1st Feb 2024

Dear Prof. Landsberger,

Thank you for the submission of your manuscript to EMBO Molecular Medicine. We have now received feedback from the three reviewers who agreed to evaluate your manuscript. As you will see from the reports below, the referees acknowledge the interest of the study and are overall supporting publication of your work pending appropriate revisions.

Addressing the reviewers' concerns in full will be necessary for further considering the manuscript in our journal, except for the proteomics and/or metabolomics studies requested by referee #1. EMBO Molecular Medicine encourages a single round of revision only and therefore, acceptance or rejection of the manuscript will depend on the completeness of your responses included in the next, final version of the manuscript. For this reason, and to save you from any frustrations in the end, I would strongly advise against returning an incomplete revision.

We are expecting your revised manuscript within three months, if you anticipate any delay, please contact us.

We require:

4) A .docx formatted letter INCLUDING the reviewers' reports and your detailed point-by-point responses to their comments. As part of the EMBO Press transparent editorial process, the point-by-point response is part of the Review Process File (RPF), which will be published alongside your paper.

5) A complete author checklist, which you can download from our author guidelines (<https://www.embopress.org/page/journal/17574684/authorguide#submissionofrevisions>). Please insert information in the checklist that is also reflected in the manuscript. The completed author checklist will also be part of the RPF.

6) It is mandatory to include a 'Data Availability' section after the Materials and Methods. Before submitting your revision, primary datasets produced in this study need to be deposited in an appropriate public database, and the accession numbers and database listed under 'Data Availability'. Please remember to provide a reviewer password if the datasets are not yet public (see <https://www.embopress.org/page/journal/17574684/authorguide#dataavailability>).

7) For data quantification: please specify the name of the statistical test used to generate error bars and P values, the number (n) of independent experiments (specify technical or biological replicates) underlying each data point and the test used to calculate p-values in each figure legend. The figure legends should contain a basic description of n, P and the test applied. Graphs must include a description of the bars and the error bars (s.d., s.e.m.). Please provide exact p values.

8) Our journal encourages inclusion of *data citations in the reference list* to directly cite datasets that were re-used and obtained from public databases. Data citations in the article text are distinct from normal bibliographical citations and should directly link to the database records from which the data can be accessed. In the main text, data citations are formatted as follows: "Data ref: Smith et al, 2001" or "Data ref: NCBI Sequence Read Archive PRJNA342805, 2017". In the Reference list, data citations must be labeled with "[DATASET]". A data reference must provide the database name, accession number/identifiers and a resolvable link to the landing page from which the data can be accessed at the end of the reference.

Further instructions are available at .

9) Author contributions: CRediT has replaced the traditional author contributions section because it offers a systematic machine readable author contributions format that allows for more effective research assessment. Please remove the Authors Contributions from the manuscript and use the free text boxes beneath each contributing author's name in our system to add specific details on the author's contribution. More information is available in our guide to authors.

10) Disclosure statement and competing interests: We updated our journal's competing interests policy in January 2022 and request authors to consider both actual and perceived competing interests. Please review the policy <https://www.embopress.org/competing-interests> and update your competing interests if necessary.

11) Every published paper now includes a 'Synopsis' to further enhance discoverability. Synopses are displayed on the journal webpage and are freely accessible to all readers. They include a short stand first (maximum of 300 characters, including space) as well as 2-5 one-sentences bullet points that summarizes the paper. Please write the bullet points to summarize the key NEW findings. They should be designed to be complementary to the abstract - i.e. not repeat the same text. We encourage inclusion of key acronyms and quantitative information (maximum of 30 words / bullet point). Please use the passive voice. Please attach these in a separate file or send them by email, we will incorporate them accordingly.

12) As part of the EMBO Publications transparent editorial process initiative (see our Editorial at <http://embomolmed.embopress.org/content/2/9/329>), EMBO Molecular Medicine will publish online a Review Process File (RPF) to accompany accepted manuscripts.

In the event of acceptance, this file will be published in conjunction with your paper and will include the anonymous referee reports, your point-by-point response and all pertinent correspondence relating to the manuscript. Let us know whether you agree with the publication of the RPF and as here, if you want to remove or not any figures from it prior to publication. Please note that the Authors checklist will be published at the end of the RPF.

I look forward to receiving your revised manuscript.

Yours sincerely,

Lise Roth

***** Reviewer's comments *****

Referee #1 (Comments on Novelty/Model System for Author):

I believe the work is excellent, but to be considered for publication, the following steps should be taken:

- Specify the number of individuals for which RNA-seq was performed.
- Conduct an in-depth study of the Differentially Expressed Genes (DEGs) within the enriched pathway for each condition. Investigate whether the same genes are deregulated across different conditions.
- Explore human, unmodified models to validate results observed in mice. Although time-consuming, it would be beneficial to specify the models for such studies.
- Perform a proteomic study under various conditions, selecting candidates and confirming that Differentially Expressed Genes (DEGs) and Differentially Expressed Proteins (PEGs) are regulated in the same direction, both upregulated and downregulated,

at both RNA and protein levels.

- Repeat RT-PCRq using an alternative housekeeping gene

Referee #1 (Remarks for Author):

Franca et al. thoroughly investigate the efficacy of neural progenitor cells (NPC) in Rett syndrome (RTT). As a result of this study, it is evident that NPC secrete factors that rescue neuronal defects in vitro (RTT neurons) and ameliorate neurological abnormalities in vivo (MECP2-deficient mice). Molecularly speaking, this study proposes that IFN-Gamma could be one of the factors responsible for this improvement. Not only that, but they also confirm that treatment with IFN-Gamma on RTT neurons or MECP2-null animals improves morphological and behavioral defects, respectively.

This work is highly exhaustive and rigorous, employing a substantial number of cellular and animal models, and it is very comprehensive. Such a comprehensive approach is essential for advancing in the treatment of RTT.

Nevertheless, I have some specific concerns:

1. The transcriptomic study conducted in WT, KO, WT+NPC, and KO+NPC animals is essential for detecting enriched pathways in these individuals. However, it is unclear how many individuals per condition were included in this study, or whether bulk RNA-seq was performed as a pool or as the average of n individuals per condition. It is important to know this information to validate the subsequent results.
2. One of the detected enriched pathways is the immune system, specifically the regulation of IFN-Gamma. It would be interesting to investigate which genes are specifically involved in these enriched pathways and whether these genes are repeated among different groups of individuals per condition.
3. Studying the genes involved in these enriched pathways could shed light on the dilemma of whether IFN-Gamma acts as a neuroprotector or neurotoxin, as described in the discussion.
4. While the results are very promising, they are less evident when using female RTT murine models compared to male models. This is a crucial factor for progressing to human treatment/trials, as not all mecp2-based regulatory mechanisms appear to be the same in mice as in humans. Therefore, do you not consider conducting tests on unmodified human cellular models? What models do you think would be most suitable?
5. Although the in vitro and in vivo results of models treated with IFN-Gamma suggest a correlation between the IFN-Gamma pathway and MECP2 alterations, it would be advisable to base these studies not only on RNA-seq (DEG) but also on proteomics (DEP) and/or metabolomics (DEM) to validate the findings more robustly.
6. The validation of transcriptomic results by PCRq-RT using Actin and Rpl13 as housekeeping genes is noteworthy. However, it is important to remember that mecp2 directly regulates actin, and using this gene as a housekeeping gene is not recommended. Another gene should be considered for this purpose.

There are a couple of typographical errors in the methodology section and figure captions:

- Cell cultures: ...removed a nd...
- Figure EV4: ...number s...

Referee #2 (Comments on Novelty/Model System for Author):

The transplantation experiments performed here are clearly dependent on a high level of expertise and are very labour intensive. My only comment is that it would have been of interest to know the impact of Mecp2 KO Neural Progenitor Cells (NPCs) in their in vitro culturing assay's? It is not essential for this story, but is certainly an experiment worth considering in the future.

Their results seems highly significant and potentially important to the the Rhett syndrome community.

Referee #2 (Remarks for Author):

The discovery that structural alterations in the X-linked methyl-CpG binding protein gene 2 (MECP2) underpinned Rhett syndrome (RS) resulted in a flood of experimental investigations as to the mechanism of action and pathways involved. Importantly, RS is regarded as a neurodevelopmental condition that becomes apparent in young females after 6-18 months of age. Symptoms include development delay, impairments in language and coordination, repetitive movements, slower growth, difficulty in walking, and a smaller head size. Naturally efforts focused on MECP2's ability to bind methylated DNA and proposed role as a transcriptional repressor. However, is now clear that MECP2 mutant mice (and other methyl-CpG binding protein genes (MECPs)) do not exhibit a transcriptional mis-expression that overlaps that of comparable hypo-methylated cells. In fact, regulation of the small number of promoter methylated genes (and repeated sequencers) present in vertebrates cannot be linked to MECPs directly, which negates the original hypothesis for MECP functions. Instead MECP2 has also been linked with roles in RNA processing, mitochondrial function, signalling pathways, cell specific regulation, severe neurobiological changes and

community effects. At the same time, it is clear from studies in mouse models that re-expression of Mecp2 in neural specific mouse KOs can rescue many aspects of early Mecerp2 deficiency. Additionally, mutation screens on a Mecerp2 KO background have identified genetic modifiers that can partially rescue aspects of loss of MeCP2 activity. These observations have encouraged the search for improved therapeutics for the condition despite the wide spectrum of symptoms.

Here the authors analyse the impact of neural transplantation with Neural Progenitor Cells (NPCs), which have the ability to ameliorate the impact of many neurodegenerative disease models and for some patients. NPCs can produce specific neurotrophic factors, reactive species and immune signalling molecules that can provide immunomodulatory or neuroprotective functions in diseased tissues.

Here, the authors results imply that RTT neurons induce WT-NPCs to secrete factors that are able to recover morphological and synaptic defects of null and heterozygous neurons of MeCP2 mutant mice.

Experimentally the authors derived WT NPCs from the subventricular zone (SVZ) of lateral ventricles isolated from 6-8 week adult female C57BL/6 mice. Using this reagent they could show by co-culture experiments that WT NPCs secrete factors which promote neuronal maturation of Mecerp2 KO primary neurons. These are carefully performed experiments, but it would've been of interest to see how MeCP2 KO NPCs behaved in this assay?

They go on to show that WT NPC transplantation for a limited time can ameliorate RTT-like symptoms in Mecerp2 deficient mice through various behavioural assays and post-poned mortality in the Mecerp2 KO mice that receive the transplanted NPCs. There was also a partial amelioration of the cognitive functions of NPC-treated KO mice.

To investigate possible mechanisms, they performed and compared RNA-seq profiles of WT and Mecerp2 KO cerebellum to reveal potential molecular pathways. Through careful bioinformatic analysis they suggest that several pathways related to RTT dysfunctions are altered in the 'rescued' mice, such as synapse organisation, ion channel transport and regulation of neurogenesis. In addition, there was an enhanced immune response, including pathways related to interferon gamma (IFN γ).

Subsequently they tested the response of cultured RTT neurons to IFN γ , which resulted in Stat1 phosphorylation and a reversion in the density and colocalization of synaptic markers in KO neurons. Injection of IFN γ into the CSF of symptomatic null animals could revert motor and memory impairments but not respiratory changes.

In conclusion the beneficial effect of NPCs transplantation and IFN γ exposure in the mouse model is highly suggestive that new therapies can be developed for RTT patients based on their investigations.

Bravo.

Referee #3 (Comments on Novelty/Model System for Author):

The manuscript is well written and easy to follow, the experiments have a logical trajectory and the results are promising in terms of suggesting beneficial effects of stem cell transplantation and IFN γ administration in RTT. Although this project has a lot of unanswered questions, the data generated is of good quality, novel and with potential high impact.

Referee #3 (Remarks for Author):

This manuscript from Frasca et al describes that co-culturing of Mecerp2 WT, Het and KO primary neurons with NPC purified from adult WT mice (in a transwell format) results in rescuing of dendrite complexity defects of Mecerp2 mutant neurons as well as in increasing synaptic density. The authors then tested the potential therapeutic effects of intrathecal transplantation of NPC into MeCP2 KO and HET mice and found that it ameliorates survival and cognitive and motor and phenotypes present in these mice. To identify pathways involved in the NPC-induced phenotypic changes, they performed bulk RNAseq in cerebellum and determined that the interferon gamma response pathway was amongst the enriched pathways. Finally, they treated cultured neurons and mice with recombinant IFN γ and saw that this treatment rescued the synaptic defect in Mecerp2 KO and Het neurons in vitro, and that treated KO mice showed amelioration of cognitive deficits.

The manuscript is well written and easy to follow, the experiments have a logical trajectory and the results are promising in terms of suggesting beneficial effects of stem cell transplantation and IFN γ administration in RTT. Although this project has a lot of unanswered questions, the data generated is of good quality, novel and with potential high impact.

There are, however, some issues that if addressed will improve the manuscript:

- What is the ratio of neuron/glia in the primary cultures? Is it similar in the WT, HET and KO cultures? These data will help in the interpretation of results in terms of the "sensing" of the NPS (do they "sense" altered cellular proportions or abnormal neurons?).
- the cells have been co-cultured for 10 days (assays) or 14 days (conditioned media). Why this different in DIV?
- the results from the BDNF western blot are not completely convincing. Fig 1I shown no difference between WT and KO cells, but an increase in WT cells cultured with NPC that is also seen in KO cells. What this indicates in the context of the NPC-induced effect is unclear to this reviewer
- It would have been interesting to test whether NPC from KO mice have a similar ameliorating effect than the observed for WT (in the context of a potential "homotypic" transplantation strategy)
- Also it appears that their secretome might be affected by soluble molecules (transwell) arising from KO cells. It would also have been interesting to analyze the phenotype of the NPC cells influenced by the KO conditioned medium.

- The impaired motor learning of the KO mice is not shown. Is the latency to fall in the first trial significantly different from the last in the WT mice but not in the KO?
- the effect in survival seem to be fully accounted for an initial delay in mortality, the curves are almost identical after the initial shift. This suggest that there is mainly an acute effect of the transplant.
- the cause for the difference in engraftment in Het vs KO mice is not discussed. Is it due to age differences, sex differences or the fact that Het mice have MeCP2 expressing and non-expressing cells?

- It is surprising that NPC transplantation does not induce transcriptional changes in WT mice.
- In Fig 6F the western blot does not seem to be representative of the data depicted in the graph. Was the data normalized to TGX-stain free only?
- rationale for not testing IFN γ administration in female HET mice, which is the most accurate model of RTT?
- Although the cerebellum seems to be justified for RNAseq experiments, it would have been informative to determine whether there are transcriptional changes in other areas of the brain (at least by QRT-PCR)

UNIVERSITÀ DEGLI STUDI DI MILANO

Reply to Reviewers

We thank all reviewers for their thorough and constructive comments. We have followed most of their observations, which, we sincerely believe permitted to significantly improve our manuscript. Please, find below a point-by-point reply.

Reviewer 1

We would like to thank the reviewer for the positive comments; we are really glad to know that the study was received as well-conducted, statistically rigorous and relevant for the field.

1. The transcriptomic study conducted in WT, KO, WT+NPC, and KO+NPC animals is essential for detecting enriched pathways in these individuals. However, it is unclear how many individuals per condition were included in this study, or whether bulk RNA-seq was performed as a pool or as the average of n individuals per condition. It is important to know this information to validate the subsequent results.

We apologize for the lack of methodological details, which we have now reported in the text, both in the section “RNA extraction, qRT-PCR, RNA Sequencing and bioinformatics” and in the legend to figure 7. Indeed, bulk-RNA sequencing was conducted on individual samples, not on a sample pool, and the sample size is now indicated.

2. One of the detected enriched pathways is the immune system, specifically the regulation of IFN-Gamma. It would be interesting to investigate which genes are specifically involved in these enriched pathways and whether these genes are repeated among different groups of individuals per condition.

We have extrapolated from the RNA-seq data the log fold changes (LFC) for each gene (15 in total) of the “Interferon-gamma response” pathway in the three relevant comparisons, i.e. KO vs WT, KO-NPCs vs KO and WT-NPCs vs WT. The results obtained indicate a general up-regulation of these genes in NPC-treated KO cortices *versus* KO, in contrast to a reduction in KO samples compared to WT. These new data are included in Figure 7, panel D.

3. Studying the genes involved in these enriched pathways could shed light on the dilemma of whether IFN-Gamma acts as a neuroprotector or neurotoxin, as described in the discussion.

We have attempted to shed light on the action of IFN γ by focusing on the function of the 15 genes in Fig.7, panel D. However, we still do not feel confident in deducing with certainty whether IFN γ acts as neuroprotective or neurotoxic. Since we have published and unpublished results showing that *Mecp2* null cells (neurons or astrocytes) poison communicating cells, we lean towards a neuroprotective action. However, at this stage, we do not feel safe to comment on this at this stage.

4. While the results are very promising, they are less evident when using female RTT murine models compared to male models. This is a crucial factor for progressing to human treatment/trials, as not all mec2-based regulatory mechanisms appear to be the same in mice as in humans. Therefore, do you not consider conducting tests on unmodified human cellular models? What models do you think would be most suitable? 4. While the results are very promising, they are less evident when using female RTT murine models compared to male models. This is a crucial factor for progressing to human treatment/trials, as not all mec2-based regulatory mechanisms appear to be the same in mice as in humans. Therefore, do you not consider conducting tests on unmodified human cellular models? What models do you think would be most suitable?

We thank the reviewer for this comment, which emphasizes the importance of including human models in preclinical studies aimed at translating a therapy into the clinics. However, considering our findings of the benefits produced by NPCs on the CNS and the involvement of the immune system, it is complicated to define a suitable and adequate *in vitro* human model that can replicate the complex system of the *in vivo* one, such as the one used in this study. An innovative possibility could be to use 3D structures that integrate human-derived neuronal cells and the immune environment, as that one described in 2023 in Nature <https://doi.org/10.1038/s41586-023-06713-1>.

5. Although the in vitro and in vivo results of models treated with IFN-Gamma suggest a correlation between the IFN-Gamma pathway and MECP2 alterations, it would be advisable to base these studies not only on RNA-seq (DEG) but also on proteomics (DEP) and/or metabolomics (DEM) to validate the findings more robustly.

We understand the suggestion. However, the requested study implies the addition of a considerable number of new animals and is certainly very expensive. Considering all this and the time needed to obtain the complete data and its analysis and validation, in agreement with the editor, we have decided to not proceed with the study. We apologize to the reviewer and hope he/she understand this.

6. The validation of transcriptomic results by PCRq-RT using Actin and Rpl13 as housekeeping genes

is noteworthy. However, it is important to remember that mecp2 directly regulates actin, and using this gene as a housekeeping gene is not recommended. Another gene should be considered for this purpose. In agreement with the reviewer, we repeated the qRT-PCR experiments using Hprt and Rpl13 as housekeeping genes. Obtained data, shown in Fig.7F, confirmed previous results. The data are shown in Fig 7F. Table 1 has been updated accordingly.

There are a couple of typographical errors in the methodology section and figure captions:

- *Cell cultures: ...removed a nd...*
- *Figure EV4: ...number s...*

We thank the reviewer for carefully reading the text, highlighting these typographical errors, which were corrected.

Referee #2 (Comments on Novelty/Model System for Author):

The transplantation experiments performed here are clearly dependent on a high level of expertise and are very labour intensive. My only comment is that it would have been of interest to know the impact of Mecp2 KO Neural Progenitor Cells (NPCs) in their in vitro culturing assay's? It is not essential for this story, but is certainly an experiment worth considering in the future. Their results seems highly significant and potentially important to the the Rhett syndrome community.

We thank the reviewer for the positive comment and for suggesting such an important experiment, which has now been included in the new Fig. 2. We found that the absence of Mecp2 does not affect the ability of NPCs to rescue synaptic defects of KO neurons. We consider this result very interesting from a therapeutic point of view.

Referee #3 (Comments on Novelty/Model System for Author):

The manuscript is well written and easy to follow, the experiments have a logical trajectory and the results are promising in terms of suggesting beneficial effects of stem cell transplantation and IFNg administration in RTT. Although this project has a lot of unanswered questions, the data generated is of good quality, novel and with potential high impact.

We are really grateful to the reviewer for the kind and encouraging words.

-What is the ratio of neuron/glia in the primary cultures? Is it similar in the WT, HET and KO cultures? These data will help in the interpretation of results in terms of the "sensing" of the NPS (do they "sense" altered cellular proportions or abnormal neurons?).

We thank the reviewer for this comment. We calculated the percentage of astrocytes, microglia and oligodendrocytes in primary cortical neurons derived from brains of WT, HET and KO embryos, reporting a total absence of Iba1+ cells, only a small presence of Olig2+ cells and ~10% of GFAP+ cells. This glia contamination is similarly present in all cultures, regardless of genotype. Data are described in Methods and shown in Fig.EV6. A brief comment on this aspect has been included in the Discussion.

-The cells have been co-cultured for 10 days (assays) or 14 days (conditioned media). Why this different in DIV?

Neurons were cultured with NPCs or NHI3T3 for 7 or 14 days, to analyse dendritic branching and synaptic density, respectively. The rationale is that synaptic puncta analyses require mature neurons (DIV>13), while the measurement of dendritic complexity, length and numbers is more easily analyzed in immature cells (DIV<8). We have better indicated these details in Methods, under the heading "Analysis of neuronal morphology and synaptic markers", and in the first part of Results.

- The results from the BDNF western blot are not completely convincing. Fig II shown no difference between WT and KO cells, but an increase in WT cells cultured with NPC that is also seen in KO cells. What this indicates in the context of the NPC-induced effect is unclear to this reviewer.

We thank the reviewer for this comment. Checking the statistical analysis, we realized that there was a significant difference between KO and WT neurons, and that NPCs treatment increases BDNF levels in both neuronal preparations, regardless of genotype. The results have been corrected in Fig.1. These observations suggest a general and intrinsic trophic support of NPCs, also evident in the analyses of dendritic branching analyses (Fig. EV1). A related comment is now included in the Results.

-It would have been interesting to test whether NPC from KO mice have a similar ameliorating effect than the observed for WT (in the context of a potential "homotypic" transplantation strategy)

The reviewer's indication is very appropriate and important, also in the therapeutic perspective of an autologous transplantation. We have performed the requested experiment and demonstrated that *Mecp2* deficiency does not alter the capacity of NPCs to rescue synaptic defects in KO neurons. The data are shown in the new Figure 2 and the results obtained are discussed.

-Also it appears that their secretome might be affected by soluble molecules (transwell) arising from

KO cells. It would also have been interesting to analyze the phenotype of the NPC cells influenced by the KO conditioned medium.

We thank reviewer for the interesting and insightful comment. In response, we used two different approaches using NPCs cultured in the absence of secretome as control, and NPCs exposed to WT or KO secretome. Analysing by bulk RNAseq the transcriptional profile of the NPCs treated with the two secretomes compared to the control, a clear effect of the secretome was revealed. However, no clear sign of a different commitment, or profound cellular alterations, were detected. We would like to use these data in a future study to identify other possible beneficial mechanisms. We also performed morphological analyses and immunostaining analyses on neurospheres treated with the different secretomes or with the control. Again, we only noticed minor differences when we compared the secretome-treated samples with the control ones. We believe that the data obtained are currently not sufficiently reliable to be included in the manuscript.

- The impaired motor learning of the KO mice is not shown. Is the latency to fall in the first trial significantly different from the last trial in the WT mice but not in the KO?

Thanks to the reviewer. We have now shown in the graph that the latency to fall in the first trial is significantly different from the last trial in WT mice (irrespective of the treatment) and in treated KO animals, according with the statistical analyses.

-The effect in survival seem to be fully accounted for an initial delay in mortality, the curves are almost identical after the initial shift. This suggest that there is mainly an acute effect of the transplant.

We thank the reviewer for this sharp observation. Interestingly, this shift positively correlates with the duration of the main grafting of NPCs. We have added this comment in the Discussion.

- The cause for the difference in engraftment in Het vs KO mice is not discussed. Is it due to age differences, sex differences or the fact that Het mice have MeCP2 expressing g and non-expressing cells?

Many data report that several factors contribute to generate a graft-friendly environment, including growth factors and cytokines that are differently deregulated in KO and Het brain. Although we initially began to explore the molecular basis of this discrepancy of grafting in the two genders and highlighted the same putative molecules of interest, such as CXCL12, we have not validated their roles. In general, we believe that the less severe neuropathological features, which characterize the HET brain compared to the KO, could play a role. We have added a sentence discussing this point.

-It is surprising that NPC transplantation does not induce transcriptional changes in WT mice.

We have carefully rechecked the bioinformatics analyses and confirm that very few DEGs emerge in the comparison WT-NPC versus WT. There could be two main reasons. First, we have other

unpublished data showing that the KO brain is particularly sensitive to treatment. In addition, there might be reduced NPC engraftment in the healthy brains of WT animals.

-In Fig 6F the western blot does not seem to be representative of the data depicted in the graph. Was the data normalized to TGX-stain free only?

We agreed with the reviewer and we repeated the WB experiment producing a novel representative image and graph.

-Rationale for not testing IFNg administration in female HET mice, which is the most accurate model of RTT?

The reviewer's comment is relevant and extends the value of the therapeutic efficacy of IFNg in RTT models. We treated Het females and analysed their short-term memory proving a rescue effect. The data have been included in figure 9.

- Although the cerebellum seems to be justified for RNAseq experiments, it would have been informative to determine whether there are transcriptional changes in other areas of the brain (at least by QRTPCR)

We thank the reviewer for this comment. In accordance with the request, we have now included in the revised version of the manuscript some information derived from the transcriptional profile performed in the hippocampus of the same WT and KO animals. The data obtained (shown in Fig EV4) indicate that the transcriptional effect is greatest in the areas closest to the grafted NPCs. Although the data are statistically weaker, a similar trend was observed.

14th Aug 2024

Dear Prof. Landsberger,

Thank you for submitting your revised study. We have now received the feedback from referee #3, who evaluated your revised manuscript, and also assessed your answers' to referee #1's comments.

As you will see below, this referee is satisfied with the revisions, and I am therefore pleased to inform you that I will be able to accept your manuscript once the following minor comments are addressed:

1/ Manuscript text:

- Please accept the previous changes and only keep in track changes mode any new modification.
- Methods: All Materials and Methods need to be described in the main text using our 'Structured Methods' format. According to this format, the Methods section includes a Reagents and Tools Table followed by a Methods and Protocols section describing the methods using a step-by-step protocol format. More information on how to adhere to this format as well as a downloadable template (.docx) for the Reagents and Tools Table can be found in our author guidelines: <https://www.embopress.org/page/journal/17574684/authorguide#structuredmethods>
- The Data Availability section should be placed after the Methods. Please note that the data should be made public before acceptance of the manuscript.
- Author contributions: CRediT has replaced the traditional author contributions section because it offers a systematic machine readable author contributions format that allows for more effective research assessment. Please remove the Authors Contributions from the manuscript and use the free text boxes beneath each contributing author's name in our system to add specific details on the author's contribution. More information is available in our guide to authors.
- Please rename "Conflict of interest" to "Disclosure statement and competing interests".
- We no longer publish a "For more information" section, please remove it from the manuscript.
- References should be listed alphabetically, with 10 authors before et al.

2/ Figures:

- We usually accommodate a maximum of 5 EV figures, do you think some of your 7 EV figures could be merged? Additionally, Figure EV7 spans two pages, but has to fit on one. If this is not possible, you could add them in an Appendix file, for which 2 pages are ok. If you choose to have an Appendix file, it should be a PDF file starting with a short table of content, and include page numbers. Appendix figures should be referred to in the main text as: "Appendix Figure S1, Appendix Figure S2" etc.
- Please make sure that all figures and figure panels are referenced in the text (currently, callouts for panels Fig. EV3, 4 and 7 are missing).
- Please address the queries from our copy editors in the figure legends:
 1. Please note that the figure EV 4d is missing in the manuscript, however the legend for the same is provided. This needs to be rectified.
 2. Please define the annotated p values $^{**}/^{***}$ as well as provide the exact p-values for the same in the legend of figure 7g; 9f-h; as appropriate.
 3. Please note that the exact p values are not provided in the legends of figures 1b-d, f-i, k; 2c-e; 3b-d; 4b-c, e-h; 6b-d; 7f; 8b-d, f; 9b-d, j; EV 1a-b, d; EV 2a-e; EV 5b.
 4. Please indicate the statistical test used for data analysis in the legends of figures 7b, d; 9f-h; EV 3a-b; Ev 4a-b; EV 5b.
 5. Please note that information related to n is missing in the legends of figures 7f; EV 1a-b; EV 5a-b.
 6. Although 'n' is provided, please describe the nature of entity for 'n' in the legends of figures 7g; EV 6.
 7. Please note that the error bars are not defined in the legends of figures EV 5a; EV 6.
 - 8 Please note that the scale bar needs to be defined for figures 8a, e.
 9. Please note that scale bar and its definition are missing for figures 5b; 6a.

3/ Checklist:

- Please check the section "Experimental animals/Animal observed in or captured from the field", as I don't think it applies to your study.
- Please fill the section "Ethics/Studies involving experimental animals".

4/ Every published paper now includes a 'Synopsis' to further enhance discoverability. Synopses are displayed on the journal webpage and are freely accessible to all readers. They include a short stand first (maximum of 300 characters, including space) as well as 2-5 one-sentences bullet points that summarizes the paper. Please write the bullet points to summarize the key NEW findings. They should be designed to be complementary to the abstract - i.e. not repeat the same text. We encourage inclusion of key acronyms and quantitative information (maximum of 30 words / bullet point). Please use the passive voice. Please attach these in a separate file or send them by email, we will incorporate them accordingly.

Please also suggest a striking image or visual abstract to illustrate your article as a PNG file 550 px wide x 300-600 px high. A cropped portion of this image will serve as thumbnail for the table of content on our webpage.

5/ As part of the EMBO Publications transparent editorial process initiative (see our Editorial at <http://embomolmed.embopress.org/content/2/9/329>), EMBO Molecular Medicine will publish online a Review Process File (RPF) to accompany accepted manuscripts.

This file will be published in conjunction with your paper and will include the anonymous referee reports, your point-by-point response and all pertinent correspondence relating to the manuscript. Let us know whether you agree with the publication of the RPF.

I look forward to receiving your revised manuscript.

Yours sincerely,

Lise Roth

***** Reviewer's comments *****

Referee #3 (Comments on Novelty/Model System for Author):

The manuscript is well written and easy to follow, the experiments have a logical trajectory and the results are promising in terms of suggesting beneficial effects of stem cell transplantation and IFN γ administration in RTT.
The data generated is of good quality, novel and with potential high impact and the reviewers critiques responded satisfactorily.

Referee #3 (Remarks for Author):

The authors were very responsive to the reviewers critics. The MS is much improved and ready for publication

The authors addressed the minor editorial issues.

4th Sep 2024

Dear Prof. Landsberger,

Thank you for sending the revised file. I am pleased to inform you that your manuscript is accepted for publication and is now being sent to our publisher to be included in the next available issue of EMBO Molecular Medicine!

Yours sincerely,

Lise Roth
